# Shortcutting from self-motion signals reveals a cognitive map in mice

Jiayun Xu[1†], Mauricio Girardi-Schappo[2†‡], Jean-Claude Beique[1,3,4], André Longtin[1,2,3,4], Leonard Maler[1,3,4]*

[1]Department of Cellular and Molecular Medicine, University of Ottawa, Ottawa, Canada; [2]Department of Physics, University of Ottawa, Ottawa, Canada; [3]Brain and Mind Institute, University of Ottawa, Ottawa, Canada; [4]Center for Neural Dynamics and Artificial Intelligence, University of Ottawa, Ottawa, Canada

## eLife Assessment

This **fundamental** work provides creative and thoughtful analysis of rodent foraging behavior and its dependence on body reference frame-based vs world reference frame-based cues. **Compelling** evidence demonstrates that a robust map, capable of supporting taking novel shortcuts, can be learned primarily if not exclusively based on self-motion cues, which has rarely if ever been reported outside of the human literature. The work, which will be of interest to a broad audience of neuroscientists, provides a rich discussion about the role of the hippocampus in supporting the behavior that should guide future neurophysiological investigations.

## Abstract
Animals navigate by learning the spatial layout of their environment. We investigated spatial learning of mice in an open maze where food was hidden in one of a hundred holes. Mice leaving from a stable entrance learned to efficiently navigate to the food without the need for landmarks. We developed a quantitative framework to reveal how the mice estimate the food location based on analyses of trajectories and active hole checks. After learning, the computed 'target estimation vector' (TEV) closely approximated the mice's route and its hole check distribution. The TEV required learning both the direction and distance of the start to food vector, and our data suggests that different learning dynamics underlie these estimates. We propose that the TEV can be precisely connected to the properties of hippocampal place cells. Finally, we provide the first demonstration that, after learning the location of two food sites, the mice took a shortcut between the sites, demonstrating that they had generated a cognitive map.

## Introduction

Animals must learn the spatial layout of their environment to successfully forage and then return home. Local (*Rosenberg et al., 2021*) or distal (*Chan et al., 2012*) landmark(s) are typically important spatial cues. Landmarks can act as simple beacons, as stimuli for response learning or for more flexible 'place' learning strategies (*Chan et al., 2012*; *Goodman, 2020*; *Nyberg et al., 2022*; *Tolman et al., 1946a*). Experimental studies addressing this issue have used one (*Collett et al., 1986*) or many (*Morris, 1981*) distal landmarks that provide allocentric coordinates for learning trajectories from a start site to a hidden location. A second source of spatial information derives from idiothetic cues that provide a reference frame via path integration of self-motion signals (*McNaughton et al., 1996*; *Mittelstaedt and Mittelstaedt, 1980*). Research on the interaction of landmark and self-motion cues has concluded that a 'spatial map' can be generated by self-motion cues anchored to stable landmarks (*McNaughton et al., 1996*; *Burgess, 2006*; *Chen et al., 2013*; *Knierim and Hamilton, 2011*).

*For correspondence:
lmaler@uottawa.ca

[†]These authors contributed equally to this work

Present address: [‡]Departamento de Física, Universidade Federal de Santa Catarina, Santa Catarina, Brazil

Competing interest: The authors declare that no competing interests exist.

This spatial map can link start sites to important locations via efficient trajectories. However, it is not clear whether such a map is sufficient for flexible navigation using entirely novel routes.

We designed a circular open maze where navigation trajectories are unconstrained, the mice's allocentric reference frame is determined by large visual cues and food is hidden in one of 100 holes. We found that mice did not use the landmarks but required only minimal cues, namely a stable start location, self-motion signals and active sensing (hole checks), to learn to find food. Trajectories and hole checking strategies dramatically changed as the mice learned to efficiently navigate to the hidden food. Averaged trajectory directions converged to a 'mean displacement direction', and hole checking became concentrated near the expected food hole. These analyses resulted in a computed target estimation vector (TEV) that closely approximated the most direct route between the start location and food. When, after learning, the mice were transferred to another start site, their TEVs also rotated and then pointed to the 'rotationally equivalent location' (REL) instead of the target, therefore demonstrating that they were utilizing self-motion cues for navigation. We propose ways to link the mean displacement and hole checking components of the TEV to the properties of hippocampal place cells and suggest a neural equivalent of the TEV that might be computed by downstream hippocampal targets.

The cognitive map hypothesis proposes that animals can learn a metric map of their environment and use it to flexibly guide their navigation, that is, they can take shortcuts, detours and novel routes when needed (*Morris, 1981*; *Tolman, 1948*; *Tolman et al., 1946b*). It is further proposed that the hippocampus and closely connected cortical areas generate a cognitive map (*McNaughton et al., 2006*; *O'Keefe and Nadel, 1978*). We found that, after training to find food in two distinct sites, mice were able to take an unrehearsed shortcut between the sites on a final probe (i.e. no food) trial. The trajectories and hole check analyses demonstrated that the computed TEVs were accurate and efficient. To the best of our knowledge, this is the first demonstration that a stable start location and self-motion cues are sufficient for a rodent to compute a cognitive map. It is not clear whether current theories based on electrophysiological studies of the hippocampus and associated cortices can account for the cognitive map we have observed (*McNaughton et al., 2006*; *Banino et al., 2018*).

## Results

We designed a behavioral framework, the Hidden Food Maze (HFM), to tease apart the roles of idiothetic and allothetic cues for navigation in a foraging task (*Figure 1A*). Mice are trained to find a food reward hidden inside one hole (the target) out of 100 in a large, circular open maze (120 cm in diameter). The target hole has a 'rotationally equivalent location' (REL) in each of the arena's quadrants (*Figure 1B*; see 'REL' in Materials and methods for a description). In the mouse's perspective, the displacement from the entrance location to the target is the same as from the entrance in any quadrant to their respective REL. The HFM has several unique features such as 90° rotational symmetry to eliminate geometric cues and control for distal cues, movable home cage entranceways, and experimenter-controlled visual cues (see Materials and methods). More importantly, our framework gives the mice freedom to explore the arena with minimum stress, yielding variable trajectories due to the nonstereotyped environment. These features allow us to control whether mice orient to allothetic or idiothetic cues, enabling us to investigate the respective role of landmarks versus path integration in spatial navigation.

We first analyzed the effects of randomizing the entrance location taken by the mouse on successive trials (*Figure 1C*), versus maintaining a static entrance throughout training (*Figure 1D*). We quantified the geometric and kinetic features of the trajectories and the behavior of hole checking throughout the arena. The variability of the trajectories resulted in the development of statistical methods to directly infer the mice estimation of the target location (the 'target estimation vector', or TEV). We showed that it coincides with the actual target vector only for the static entrance protocol. In order to eliminate the effect of visual cues, we performed rotated probe trials after training in static entrance, where mice went either to the target (following landmarks) or to the REL (ignoring landmarks; see *Figure 1D*).

Finally, we investigated the features of the mice trajectories and active sensing behavior when trained on two targets consecutively under the static entrance protocol (*Figure 1E*). In this case, the TEV method was employed to demonstrate the learning of both target directions and the use of

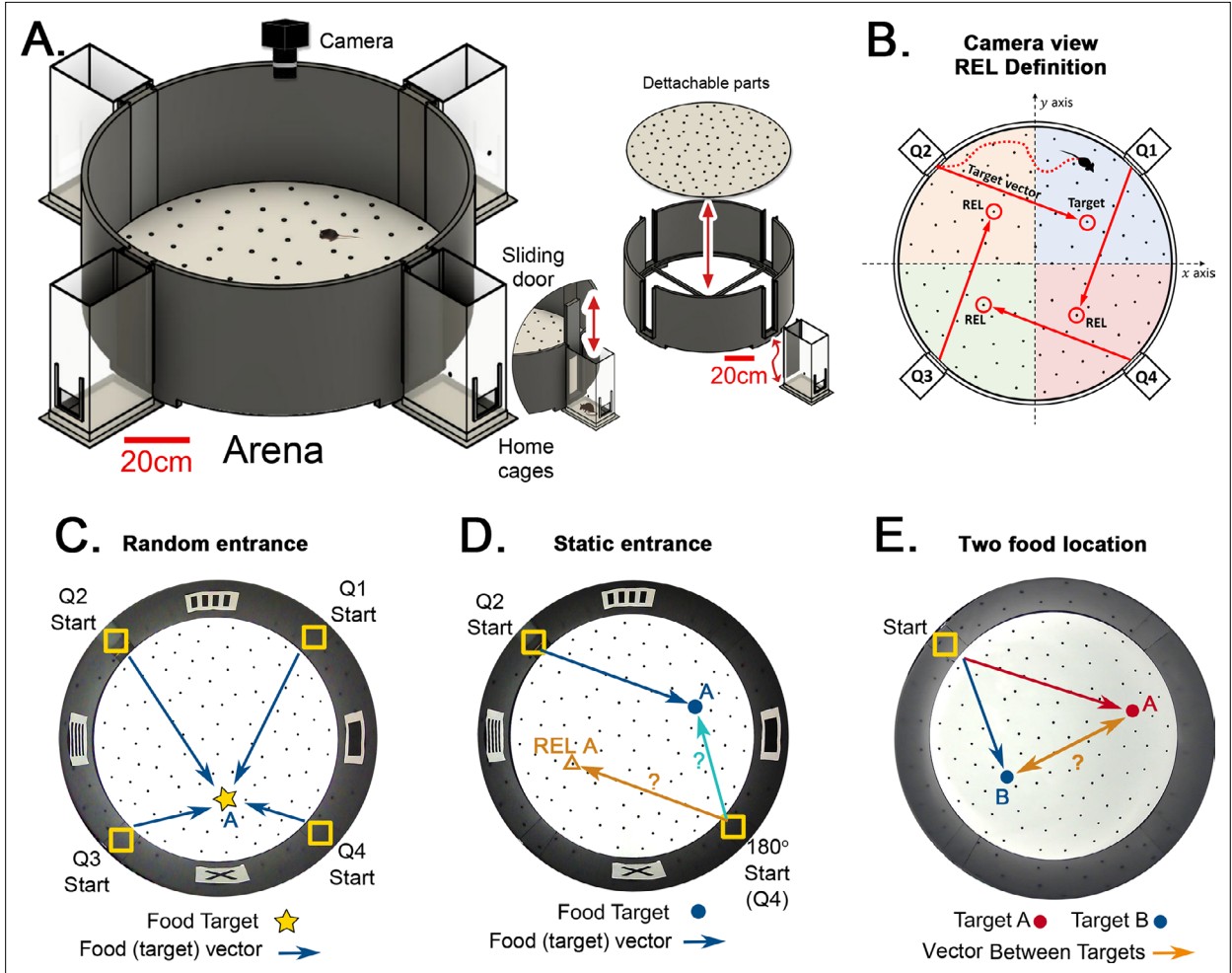

**Figure 1.** Hidden Food Maze and experimental setup. (**A**) The floor of the arena is 120 cm in diameter, and the walls are 45 cm tall. Note 20 cm scale bar in this panel. The home cage has a 10.5x6.5 cm² floor area. The door slides upward (mice enter without handling). The floor is washed and rotated between every trial to avoid predictable scratch marks and odor trails. The subfloor containing food is not illustrated. (**B**) Camera view of a mouse searching for hidden food (target, pointed by the target vector). The REL of the target is marked for each entrance (from the mouse's perspective, the displacement from the start in each quadrant to its respective REL is the same for any entrance; e.g. '70cm forward +30 cm to the left'; and it is equal to the displacement from the trained start to target). (**C**) 'Random entrances' experiment. Mice enter from any of the four entrances randomly over trials to search for food ('A'-labeled star) always in front of the X landmark. Arrows show the four possible displacements. (**D**) 'Static entrances' experiment. Mice start from the same entrance (labeled 'Start') in every trial to search for food in front of the same landmark. Blue/cyan arrows = food vector (start→food); Orange arrow = REL vector (start→REL). After training, the start position in a probe trial can be rotated ('180° Start') to check whether mice follow idiothetic (start→REL; ignoring landmarks) or allothetic (start→A; following landmarks) cues; going via the REL vector is regarded as evidence of path integration. (**E**) 'Two food location' experiment. Mice start from a static entrance to search for food (red vector to target A). Afterwards, mice are trained to find food in a different location (blue vector to target B). After learning both targets, a probe trial (i.e. a trial without food) is designed to check whether mice can compute shortcuts from B to A (B-A vector, orange arrow).

The online version of this article includes the following figure supplement(s) for figure 1:

**Figure supplement 1.** Pre-training trajectories and active sensing from the mouse perspective in random entrance experiment.

shortcuts, one manifestation of a cognitive map (***Tolman et al., 1946b***). See Materials and methods and the figure supplements for the definition of all measured quantities and details on experimental protocols.

## Pretraining trials

Initially, naïve mice were introduced to our maze. In agreement with previous work (***Fonio et al., 2009***), we observed that mice initially explored the maze by following the wall (thigmotaxis) on round trips of increasing complexity before they explored the maze center. Our pretraining protocol induced

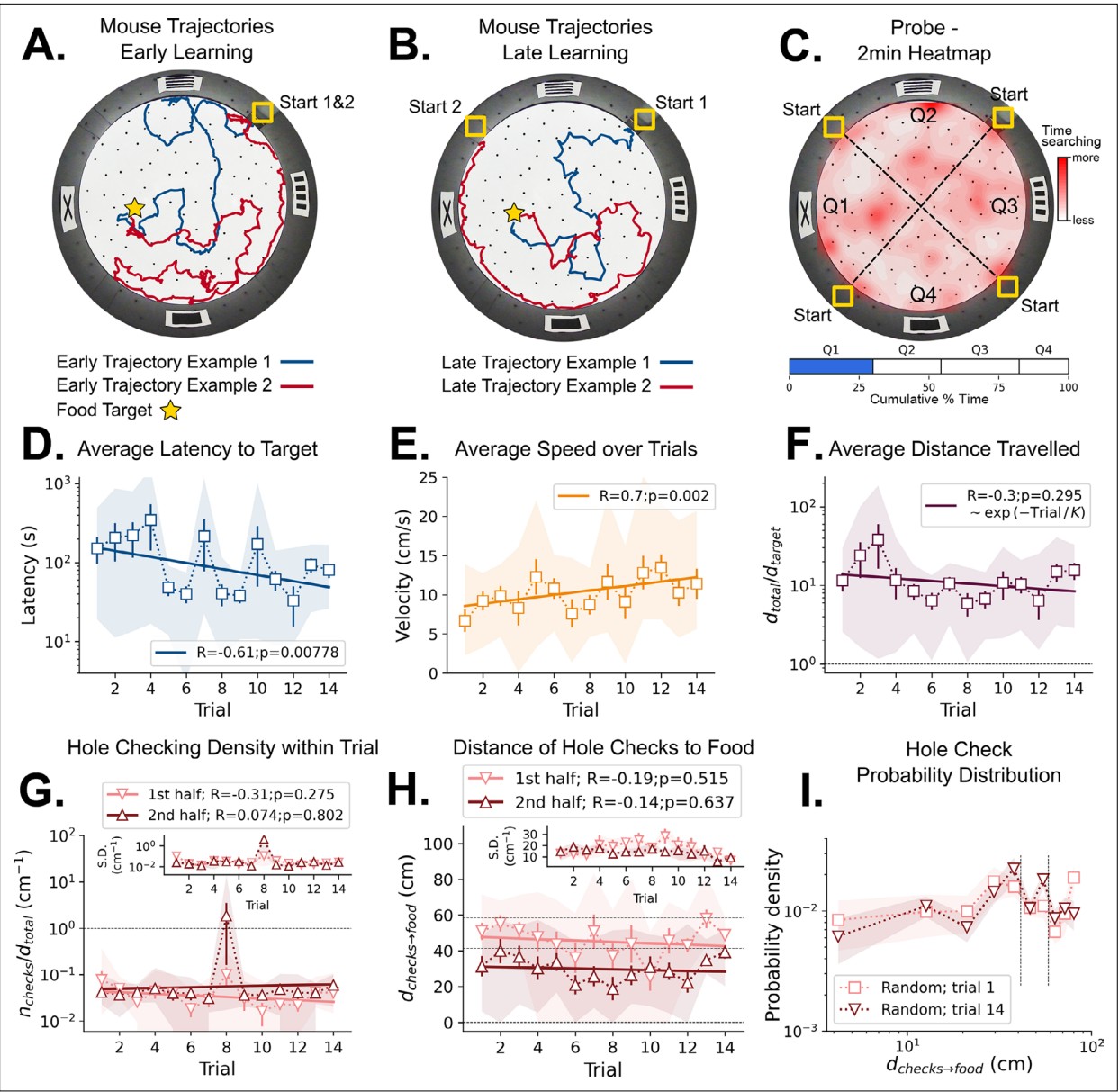

**Figure 2.** Mouse spatial learning with random entrances. (**A**) Two examples of mouse search trajectories during early learning (trial 3) when the entrance changes from trial to trial. They are irregular and vary unpredictably across trials. (**A, B**) Star = target location. Yellow Square = entrance site. (**B**) Two examples of mouse search trajectories during late learning (trial 14) after starting from different entrances. Trajectories look as irregular as in early trials. (**C**) Top: Heatmap of the first 2 min of a probe trial done after trial 18 (red = more time in a given region). **Bottom:** Mice spent about the same time (25%) in each of the four sectors, regardless of being close to the target (blue) or to its REL (white). (**D**) Some significant reduction in latency to reach target is seen across trials (p=0.008; N=8). (**D–I**) Error bars = S.E. Shaded area = data range. (**E**) Some significant increase in speed is seen (p=0.002; N=8). (**F**) Average normalized distance traveled to reach target ($d_{total}/d_{target}$=1 is optimal; p=0.05, N=8). (**G**) Hole-checking density (number of hole checks per distance traveled) in each half of the trajectory. The density remains constant for both halves and across trials, suggesting that mice remained uncertain as to the food location. G-inset. The S.D. of the density over the mice sample remains constant for both halves (N=8). (**H**) The average distance of the checked holes to the food $d_{(checks \to food)}$ remains almost constant across trials. Horizontal lines are just guides to the eye. (**I**) The probability density of the distance of hole checks to the food $d_{(checks \to food)}$ for the first and last learning trials (the corresponding averages over trials are in panel **H**). The density remains unaltered. Vertical dotted lines mark the same distances as the horizontal lines in panel (**H**).

The online version of this article includes the following figure supplement(s) for figure 2:

**Figure supplement 1.** Hole check distribution in Random Entrance protocol for the four target holes from the mouse's perspective.

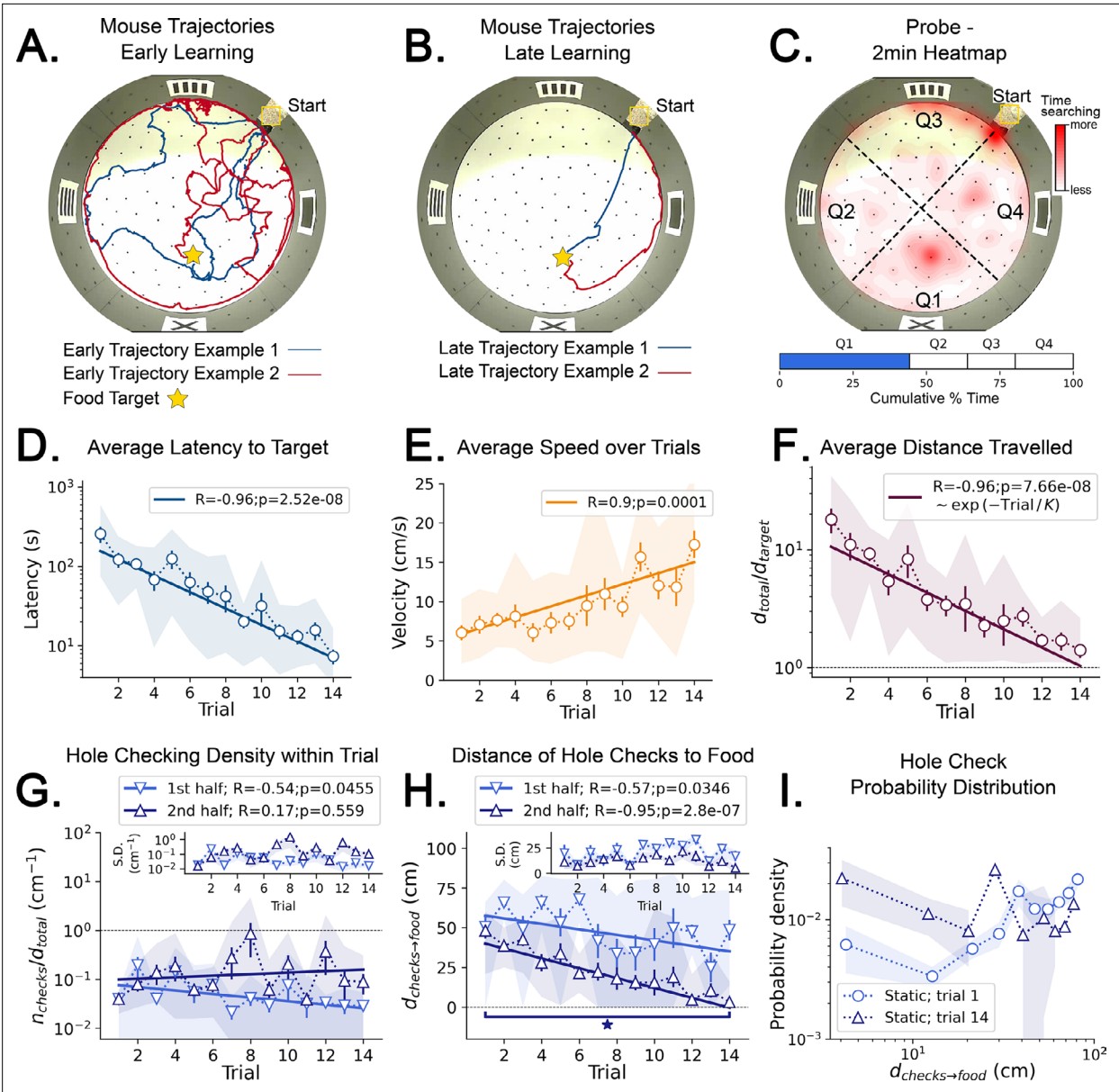

**Figure 3.** Mouse spatial learning with static entrances. (**A**) Two examples of mouse search trajectories during early learning (trial 3). They are irregular and variable similarly to those in the random entrance experiments. (**A, B**) Star = target location. Yellow Square = entrance site. (**B**) Two examples of mouse search trajectories during late learning (trial 14). They go directly towards the food or go along the wall before turning to the food, creating variation across mice and trials. (**C**) (Top) Heatmap of the first 2 min of a probe trial done after trial 14 (red = more time in a given region). (Bottom) Mice spent almost 50% of the time within 15 cm radius of the target (blue) compared to the RELs (white). (**D**) Latency dramatically decreases ($p < 10^{-7}$; N=8). (**D–I**) Error bars = S.E. Shaded area = data range. (**E**) Speed significantly increases during trials (p=0.0001; N=8). (**F**) Normalized distance to reach target $d_{(total \to target)}$=1 is optimal becomes almost optimal ($p < 10^{-7}$; N=8). (**G**) Hole-checking density over distance in each half of the trajectory. It significantly decreases in the first half (p=0.05), and stays constant in the second. G-inset: The S.D. of the density is larger in the second half. (**H**) The average distance of the checked holes to the food $d_{(checks \to food)}$ decreases for both halves of the trajectory. After learning, the hole checks happen closer to the food $d_{(checks \to food)}$ is almost zero, although there are more checks per distance. (**I**) The probability density of the distance of hole checks to the food $d_{(checks \to food)}$ for the first and last trials (the corresponding averages over trials are in panel **H**). After learning (trial 14), the density is larger closer to the food, a feature that does not appear in the random entrance experiments.

The online version of this article includes the following figure supplement(s) for figure 3:

**Figure supplement 1.** Location of hole checks in the last 3 s before finding the target in Static Entrance protocol.

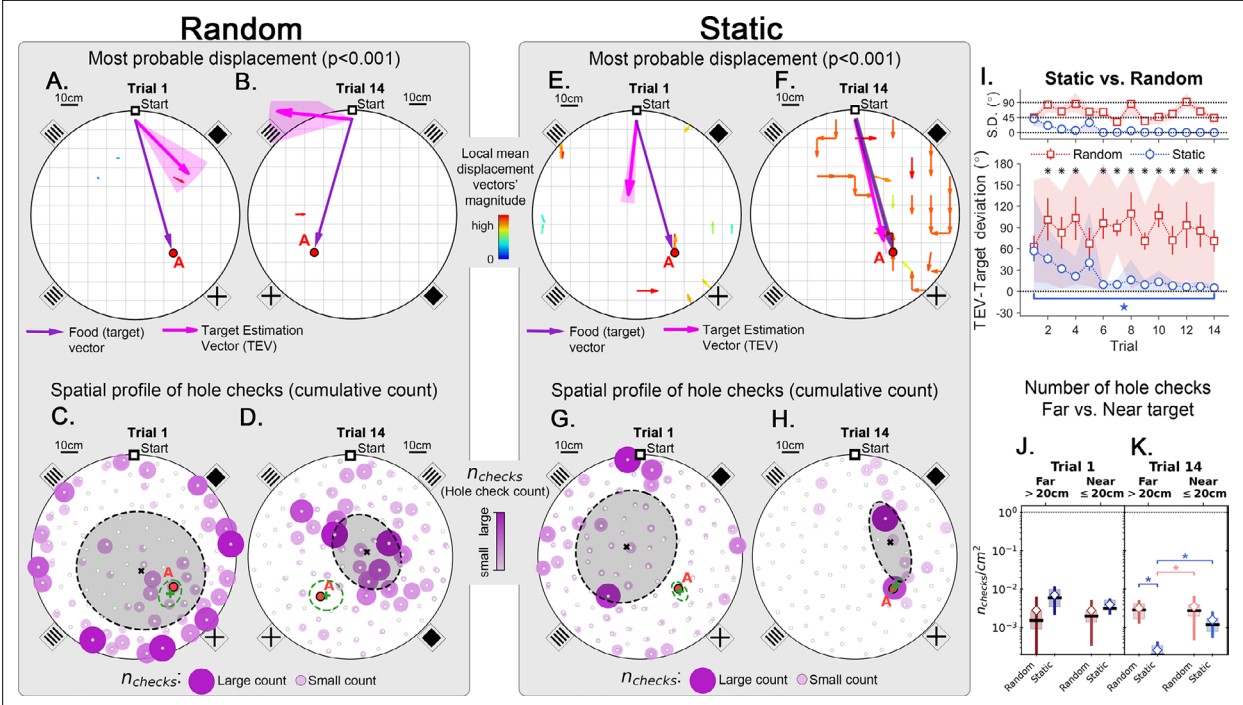

**Figure 4.** Trajectory directionality and active sensing for random and static experiments. Arenas on the top row (mean displacement vector – see color scale between panels **B** and **E**) correspond to the ones immediately below them (hole checking spatial distribution); the red 'A' label marks the target (food site), which is pointed by the food (target) vector (purple arrow). Top row (**A, B, E, F**): the color and arrows indicate the most probable route taken (red = more probable; only p<0.001 displacements shown; pink arrow = inferred target position, or TEV; shaded pink sector = S.D. of TEV; see Materials and methods, and *Figure 6—figure supplement 1*). Bottom row (**C, D, G, H**): spatial distribution of hole-checks; size and color of circles = normalized frequency that a hole was checked (larger pink circles = higher frequency); Black ellipse (x=mean): covariance of spatial distribution. Green ellipse (+=mean): covariance of spatial distribution restricted to ≤20 cm of the target. Random entrance experiments (N=8; panels **A, C**: trial 1; **B, D**: trial 14): regardless of training stage, no significant preferred routes and the TEV does not point to target (**A, B**); hole checks are randomly distributed throughout the arena, and shift from the walls (**C**) to near the center (**D**) after learning. Static entrance experiments (N=8; panels **E, G**: trial 1; **F, H**: trial 14): after learning (**F**) the TEV and significant displacements go straight to the target (although individual trajectories are variable); and hole checks align along the start-target path (**H**). (**I**) Deviation between the TEV (pink arrow) and the target vector (purple arrow) illustrated in top panels. Directionality is quickly learned (static case). (**J, K**) Hole-check area density corresponding to the spatial profiles in bottom panels. Density after learning is larger near the target (static case), supporting the path integration hypothesis. Asterisks/star: p<0.05 (paired t-test). Note the presence of more significant displacements in late learning for static entrances only, and the associated alignment of the TEV and food vector.

The online version of this article includes the following figure supplement(s) for figure 4:

**Figure supplement 1.** Definition of a hole-checking event as active sensing and target estimation vector (TEV).

**Figure supplement 2.** Estimation of significance for the mean displacement direction calculation.

**Figure supplement 3.** Kinetic and geometric features and correlation with active sensing.

**Figure supplement 4.** Covariance between geometric and kinetic features, and active sensing.

**Figure supplement 5.** Control experiments for path-integration.

the mice to explore the arena center (see *Figure 1—figure supplement 1A*, Materials and methods). Thigmotaxis was still observed after pretraining (see, e.g. *Figures 2A, B ,, 3A and B*), although now the mice spent more time in the central regions of the maze. In the Static trial experiments, there was not a distinct class of thigmotactic trajectories, but rather a continuum of trajectory distances from the wall. Furthermore, the mice were initially equally likely to take trajectories at any angle from the start-to-food line as evidenced by the variation of the TEV deviation from the target vector in trial 1 of *Figure 4I*.

## Mice used self-motion sensory input but did not utilize 2D wall cues for spatial learning

### Random Entrance experiments

Mice were trained with their cage being moved randomly between quadrant entrances from trial to trial without directly manipulating the animals. Thus, they never entered from the same start position for consecutive trials, making the wall pictures their only reliable orienting landmarks. These cues were 15x15 cm black shapes on a white background (see Materials and methods). The configuration of the 4 cues varied relative to the entrance, but the food was always placed in the same hole and nearest the X-shaped landmark (*Figure 2*, Materials and methods). The food-restricted mice showed persistent complex search trajectories throughout the learning period (*Figure 2A and B*). The latency to reach the food target was significantly negatively correlated with trial number (statistics in *Figure 2D*) but with a non-significant difference between the first and last trials (N=8; First trial mean ± SD = 151.34±148.91 s; Last trial mean ± SD = 80.38±43.86 s; p=0.2602). Despite the decrease in latency, mice did not appear to exhibit spatial search strategies (*Figure 2C*) given that the time spent in each quadrant during the probe trial is similar (target quadrant = 28.78%, other quadrants = 24.93%, 28.04%, 18.25%; one-way ANOVA: p=0.070574, F-ratio=2.71621; see Behavioral analysis in Materials and methods). This suggests latency decreased for reasons independent of spatial learning.

We found a corresponding significant positive correlation of speed and trial number (*Figure 2E*) with non-significant differences between first and last trials (N=8; First trial mean ± SD = 6.68±3.79 cm/s; Last trial mean ± SD = 11.40±5.07 cm/s; p=0.0704). Since the distance from start to target ($d_{target}$) varies over trials due to changing entrance location, we defined the normalized trajectory length (*Figure 2F*). It is the ratio between total traveled distance ($d_{total}$) and $d_{target}$; $d_{total}/d_{target} = 1$ is optimal. There is no significant correlation of this quantity and trial number (p=0.295). The first and last normalized trajectory lengths were not significantly different either (N=8; First trial mean ± SD = 11.79±8.25; Last trial mean ± SD = 15.51±9.92; p=0.4241).

Fitting the function $d_{total} = B*\exp(-\text{Trial}/K)$ reveals the characteristic timescale of learning, $K$, in trial units (*Figure 2F*). We obtained K=26 ± 24 giving a coefficient of variation (CV) of 0.92. The mean, K=26, is therefore very uncertain and far greater than the actual number of trials. Thus, we hypothesize that the mice did not significantly reduce their distance travelled (*Figure 2A, B and F*) because they had not learned the food location – the decrease in latency (*Figure 2D*) was due to its increased running speed and familiarity with non-spatial task parameters.

We next examined hole checking as an active sensing indicator of the mouse's expectation of the food location (see Materials and methods, *Figure 4—figure supplements 1–4* and the Shortcut Video) for hole-check detection after spatial learning. Even though hole check counts increase with the total trajectory length and latency (*Figure 4—figure supplement 3B and C*), we expected that the mice would increase the number of hole checks near the expected food location, or along their path toward it. Thus, we split the trajectories into two halves of same duration to identify whether hole checks would increase in the second half (being closer to the target). However, the hole checking density (ratio between number of checks and distance travelled) remains constant between the first and second halves of any trial (*Figure 2G*), suggesting mice did not anticipate where the food is placed. Mice also did not increase their hole sampling rate as they approached the target as shown by the distance of hole checks to the target, $d_{checks \to food}$ (*Figure 2H, I*), again suggesting no expectation of its location. Mice do not appear to learn a landmark-based allocentric spatial map.

### Static entrance experiments

We tested whether the mice might acquire an allocentric spatial map if, contrary to the random entrance case, they had a single stable view of the landmarks and could associate one cue configuration with the food location (see Materials and methods). For this, each mouse entered the arena from the same entrance during the training trials (hence 'static entrance'). Mice greatly improved their trajectory efficiency after training (*Figure 3A and B*), with distance and latency to food plateauing around trial 7. During early learning, mice showed random search trajectories that transformed to stereotyped trajectories towards food by late learning (*Figure 3B*). Typical trajectories were either (i) directly to the food, (ii) initially displaced towards the maze center then returning to a food-oriented trajectory, or (iii) initially along the wall and then turning and heading to the food (*Figure 3B*).

During the probe trial, mice spent the most time searching in the target quadrant (44.26%; *Figure 3C*), compared to the time spent in the other quadrants, respectively 19.38% (p=0.007), 16.55% (p=0.0065), 19.81% (p=0.007; p-values obtained for a pairwise t-test and adjusted for multiple comparisons via the Benjamini-Hochberg method). The latency to target strongly decreased over trials (*Figure 3D*) and the last trial latency was significantly reduced compared to the first (N=8; First trial mean ± SD = 255.39±153.28 s; Last trial mean ± SD = 7.38±4.03 s; p=0.004). The speed dramatically increased over trials (*Figure 3E*) and the ending speed was significantly greater than the initial speed (N=8, First trial mean ± SD = 6.04±2.48 cm/s; Last trial mean ± SD = 17.25±4.64 cm/s; p=0.0002). The normalized trajectory length strongly negatively correlated with trial number (*Figure 3F*) and the final normalized length greatly reduced (N=8; First trial mean ± SD = 18.02±11.36; Last trial mean ± SD = 1.40±0.50; p=0.0061), becoming almost optimal (*Rosenberg et al., 2021*).

Now the fitting of the function $d_{total} = B \exp(-\text{Trial}/K)$ yielded $K=5.6 \pm 0.5$ with a CV = 0.08; the mean is therefore a reliable estimate of total distance travelled. We interpret this to indicate that it takes a minimum number of K=6 trials for learning the distance to the target (see also *Figure 4—figure supplement 3D, E, F and G*). Learning is still not complete because it takes 14 trials before the trajectories become near optimal.

Hole checking density varied across trajectories and differed between early and late learning trials. The density significantly decreased with trial number in the first half of trajectories ($R=-0.54$; p=0.0455, *Figure 3G*) but not the second half ($R=0.17$; p=0.559, *Figure 3G*). The hole checks also happened significantly closer to the target in the second half of the trajectory (*Figure 3H*; First trial mean ± SD = 48.06±11.50 cm; Last trial mean ± SD = 3.13±5.70 cm; p=3 × 10⁻⁶). A remarkable effect of learning is that the probability of checking holes increases as the mouse approaches the food (*Figure 3I*). These results (*Figure 3H, I*) suggest that the mice may be predicting the food location and checking their prediction as they approach the estimated food location (see also *Figure 4—figure supplements 3 and 4* for other quantities and scatter plots of trajectory features vs. hole checking). Hole checking near the food site also implies that the mice are not using odorant or visual cues to sense the precise food location, but instead rely on a memory-based estimate of the food-containing hole (see also *Figure 4G and H*). This motivated us to look at the spatial configuration of these variables to get a detailed picture of the random vs. static condition.

## Revealing the mouse estimate of target position from behavior

Although quantitative, the analysis so far only showed a broad comparison between random versus static entrance experimental conditions. We developed a method to map out the directions that the mice stepped towards more frequently from each particular location in the arena, yielding a *displacement map* (see Statistical analysis of trajectories in Materials and methods). This map can generate a precise estimate of the mean directionality of the trajectories for each experimental condition (*Figure 4A, B, E, F* and p-values in *Figure 6—figure supplement 1*). Additionally, we also made a frequency plot of the checks for each particular hole in the arena, yielding the spatial distribution of hole checks (*Figure 4C, D, G and H*). These two maps are combined to give the Target Estimation Vector (TEV). The TEV is interpreted as the mouse's estimate of the target position (pink vectors in *Figure 4A, B, E and F*).

In random entrance training, no significant directionality was found in the arena (*Figure 4A and B*), again consistent with a random search. The spatial distribution of hole checks migrated from near the walls towards the center of the arena but remained random and displaced from the target (*Figure 4C and D*, and also *Figure 2—figure supplement 1*), consistent with the broad results of *Figure 2*. Conversely, there was a clear and significant flow of trajectories for mice trained with static entrance (*Figure 4E and F*; p<0.0001; see Materials and methods for the definition of the p-values assigned to directions). The hole checks reflected this and became solely concentrated along the route from start to target (*Figure 4G and H*). In the static entrance condition, the entropy (uncertainty) of the number of hole checks near (<20 cm) the target was lower than that for the far (>20 cm) condition (*Figure 6—figure supplement 1*). In other words, mice were more consistent in the number of their hole checks near the target compared to far from the target, suggesting that they had an internal representation of their proximity to the food site.

We compared the deviation between the TEV and the true target vector (that points from start directly to the food hole; *Figure 4I*). While the random entrance mice had a persistent deviation

between TEV and target of more than 70°, the static entrance mice were able to learn the direction of the target almost perfectly by trial 6 (TEV-target deviation in first trial mean ± SD = 57.27° ± 41.61°; last trial mean ± SD = 5.16° ± 0.20°; p=0.0166). A minimum of 6 trials is sufficient for learning both the direction and distance to food (*Figure 4I*) (*Figure 3F*) (see Discussion). The kinetics of learning direction to food are clearly different from learning distance to food since the direction to food remains stable after Trial 6 while the distance to food continues to approach the optimal value.

Under the path integration hypothesis, it is expected that error accumulates as the mouse walks (*McNaughton et al., 2006*). This motivated us to investigate the frequency of hole checks per area near the target (within 20 cm) vs. far (further from 20 cm away), *Figure 4J and K*. The density of hole checks in both conditions remained constant for random entrance mice, regardless of training duration. However, static entrance mice had significantly higher density of hole checks near the target (*Figure 4K*; p-values: random-far vs. static-far=0.005; static-far vs. random-near=0.03; random-near vs. static-near=0.01). The mean position of hole checks near (≤20 cm) the target is interpreted as the mouse estimated target (*Figure 4C, D, G and H*; green + sign = mean position; green ellipses = covariance of spatial hole check distribution restricted to 20 cm near the target). This finding together with the displacement and spatial hole check maps (*Figure 4F and H*, respectively) corroborates the heatmap of time spent in the target quadrant (*Figure 3C*), suggesting a positive correlation between hole checks and time searching (see also *Figure 4—figure supplement 3C*). Thus, we use the distance from this point to the entrance as the magnitude of the TEV. This definition reveals that the TEV is nearly coincident with the direct route between start location and the food-containing hole (pink vectors in *Figure 4F*), consistent with the near optimal normalized trajectory length previously discussed (*Figure 3F*).

## Ruling out landmarks

These results demonstrated that mice can learn the spatial location of the food target from a static entrance. We next investigated whether the performance gain can be attributed to the cues by manipulating the mouse's entrance. Well-trained static entrance mice were subject to rotation of their home cage by 180°, +90°, or –90° for probe trials in order to start from a different location. We used a different cohort of N=8 mice for each rotation. The mice were not directly handled during the rotation procedure. If the mice were using the distal landmarks, they would still be expected to find the correct food location. If they relied on self-motion cues, they should travel to the REL, since this is the spot where the target would have been if the mouse had been trained from that entrance. In other words, going to the REL means that the trajectory is anchored to the mouse's start point and not to the wall cues. As illustrated in *Figure 1B*, the displacement between each start-REL pair is exactly equal from the mouse's point of view (e.g. going 70 cm forward then 30 cm to the left reaches the respective REL of the target).

Following all rotations, mice searched at the REL instead of the correct location for the food reward regardless of the arena having wall cues (*Figure 5A, B and C*), meaning that they were not using landmarks to compensate for rotation. In fact, the trajectory kinetics of the 180°-rotated probe for reaching the REL target (mean ± SD latency = 4.80 ± 3.06 s; speed = 14.60 ± 5.21 cm/s; normalized distance to REL = 1.96 ± 0.90) is indistinguishable (within one SD or less) from the kinetics of the last trial of static entrance for reaching the true target. The criterion for reaching the REL target was getting within 5 cm of its position (the average inter-hole distance is 10 cm).

We also show the displacement maps (mean trajectory directionality), hole check spatial profile and TEV for the 180°-rotated probe trial (*Figure 5C*). The TEV now points to the REL, and the hole checks accumulate along the route to the REL. These maps are also very similar to the static entrance maps in *Figure 4*. Thus, the mice failed to use allothetic cues for navigation in the novel start location test.

## Further controls

It is possible that mice, like gerbils (*Collett et al., 1986*), choose one landmark and learn the food location relative to that landmark (e.g. the 'X' in *Figure 3B*); in this case, the mice would only have to associate their initial orientation to that landmark without identifying the image on it. We used two additional controls to assess this possibility. Mice were trained without any cues on the wall, eliminating the possibility that they provided orienting guides. The learning curve of mice with or without the 2D wall cues were not significantly different (*Figure 4—figure supplement 5*).

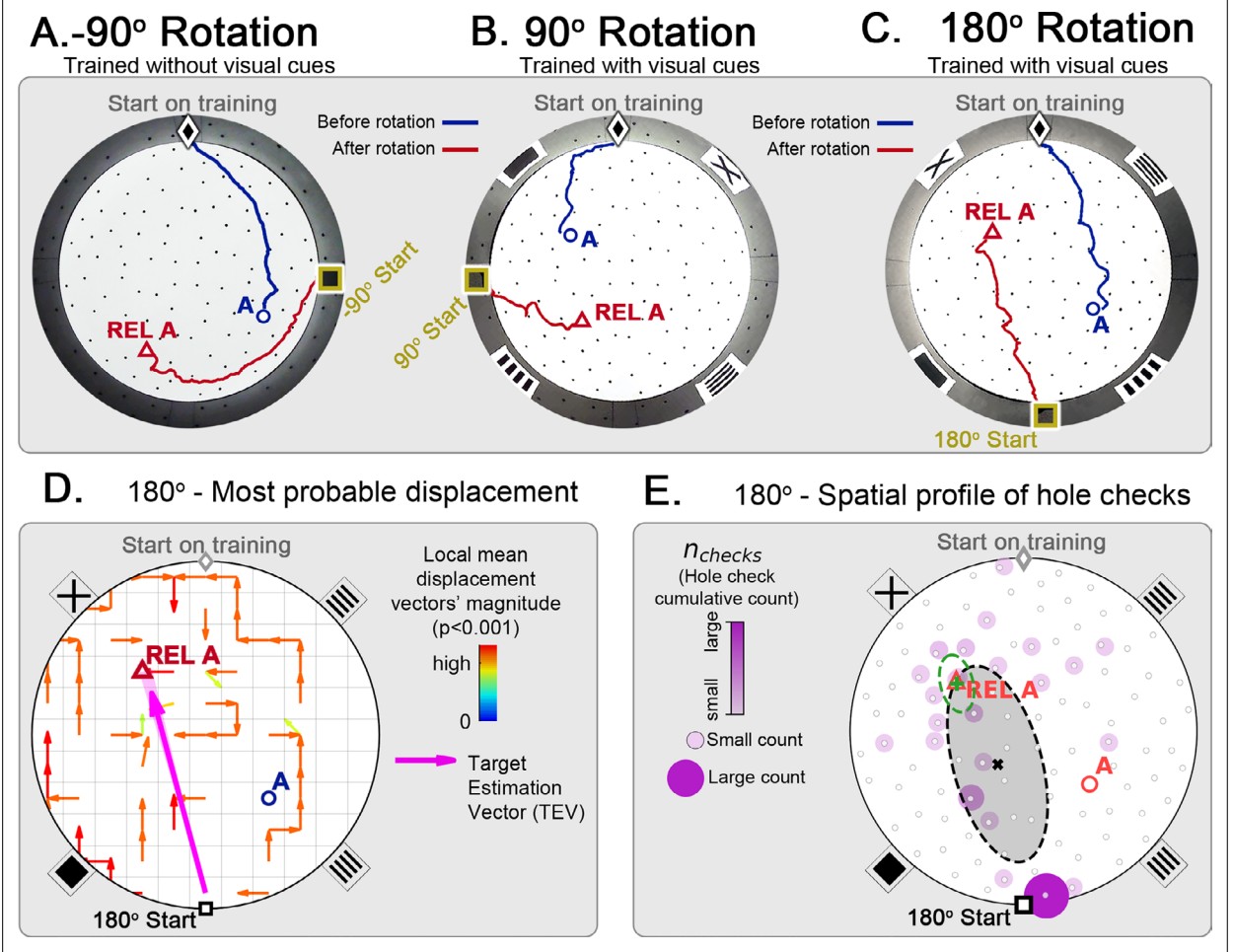

**Figure 5.** Changing start position after training in static protocol. Mice are trained in the static entrance protocol to find food at the target labeled 'A' (blue circle), and a probe trial is executed with mice entering from a rotated entrance after 18 trials. (**A, B, C**) show the comparison between trajectories from the last learning trial (blue) versus the probe (red). The training was performed without landmarks (**A**) N=8, –90° rotation and with landmarks (**B**) N=8, 90° rotation; (**C**) N=8, 180° rotation. In all instances, mice ignored landmarks and went to the REL location ("REL A" label, red triangle), something that is expected under the path integration hypothesis. (**D**) Trajectory directionality analysis and TEV (pink arrow; shaded sector: S.D.) show that significant paths of all mice (p<0.001; N=8; see Materials and methods) point to the REL-A location in the same way that it pointed to the target without rotated entrance in *Figure 4F*. (**E**) The spatial distribution shows that hole checks accumulate along the start-REL vector, instead of the start-target vector of the case without rotation in *Figure 4H*. Black ellipse (x=mean): covariance of hole check distribution. Green ellipse (+=mean): covariance of the data within 20 cm of the REL-A location. This suggests that mice follow trajectories anchored to their start location (idiothetic frame of reference).

Mice that were well trained in the lighted maze were given trials in complete darkness. These mice still showed the same learning curve compared to untrained mice or mice following rotation (*Figure 4—figure supplement 5*).

Finally, to completely rule out both landmark and possible olfactory cues, we trained and tested mice in total darkness. Head direction tuning may be impaired in sighted mice navigating in darkness (*Asumbisa et al., 2022*), but spatial learning was not affected in our experiments. Mice can also use localized odor sources as landmarks for spatial learning (*Fischler-Ruiz et al., 2021*) and we therefore attempted to eliminate such cues (see Discussion and Materials and methods). The mice were still able to learn the food location in the absence of visual and olfactory cues (*Figure 4*, *Figure 4—figure supplement 5*). These experiments demonstrate that the mice can learn to efficiently navigate from a fixed start location to the food location using path integration of self-motion cues.

Furthermore, the location of the targets in all instances of the experiment was carefully chosen to avoid cues from the arena's circular geometry. If the target was placed near the center of the arena, the learning task would be trivial and would involve little spatial inference. Conversely, if the target

was placed too close to the wall, thigmotaxis would be sufficient to get near the food. Also, the line going through the center of the arena that is perpendicular to the line connecting start to center has to be avoided (see *Figure 1B* for reference), otherwise the system would be symmetric to rotation or could be simplified to a left-or-right choice that would involve little spatial learning. Finally, the target cannot be placed near the start positions. Thus, in the static entrance experiments, the chosen target position is always more than 35 cm away from the wall, and more than 60 cm away from the start. In the random entrance case, the target is placed more than 30 cm away from the wall, and more than 40 cm away from the closest starting location.

## Mice can use a shortcut to navigate between remembered targets

Our results established that mice learn the spatial location of a food reward using path integration of self-motion cues. Have the mice learned a flexible cognitive map? A test of cognitive mapping would be if mice could take a short cut and navigate from one remembered location to another along

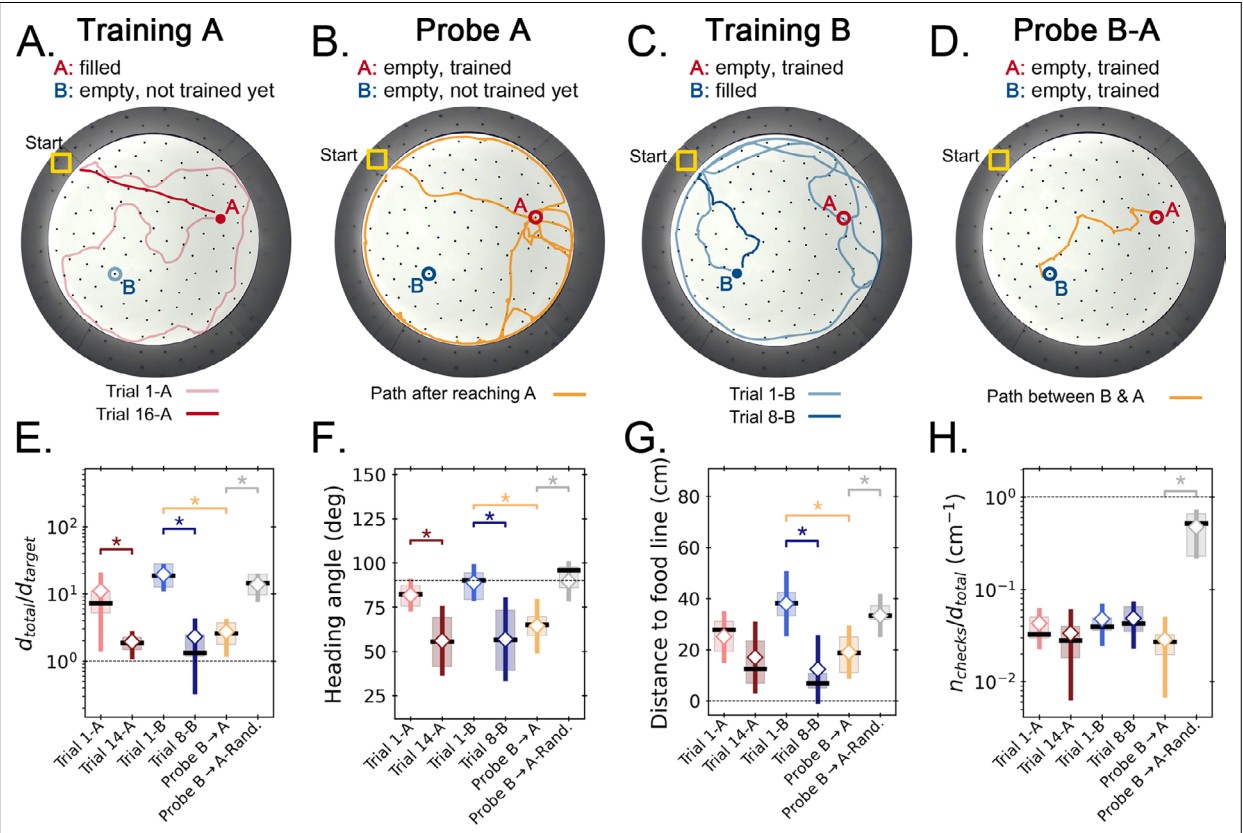

**Figure 6.** Two food location experiment. (A-D) Trajectory exemplars of four sequential stages of the experiment (all trials done with static entrance, N=8): (A) training target A (trials 1 A and 16 A for early and late learning, respectively); (B) probe A (no food is found, triggering a random search); (C) training target B (keeping A empty; trials 1-B and 8-B for early and late learning); (D) probe B-A (where both targets are trained and empty, and the mice take a shortcut from B to A; see *Figure 2—figure supplement 1* for all exemplars). Filled circles = filled target hole; empty circles = empty target. In the A and B learning stages, the trajectories evolve from random to going straight from start to the respective target. (E,F) Standard boxplot statistics of learning versus probe (diamonds are averages; asterisks: p<0.05 in a paired t-test comparison). Quantities are defined in *Figure 4—figure supplement 3A*. Significant differences between early and late learning were observed for the traveled distance (E), heading angle (F), and distance to the food line (G). Density of hole checks (H) remained nearly constant, as expected. In all instances, the values of all quantities in the B-A probe resembled the values of the late learning trials, whereas the randomized B-A probe (gray) had values that resembled early learning, suggesting the B-A behavior is not random. In the Probe B→A trials, the 'food line' is the straight line that connects B to A, along which the reference distance $d_{target}$ is measured between A and B. In the other trials, the food line is a straight line from start to the specific target, either A or B, along which $d_{target}$ is measured between start and target.

The online version of this article includes the following figure supplement(s) for figure 6:

**Figure supplement 1.** Target estimation vector (TEV) in detail for the random, static and two-target experiments.

**Figure supplement 2.** Minimum distance to alternative target in 2-target condition, and short cut trajectories.

a novel, unreinforced path. The 'two food location experiment' set out to test this possibility by training mice to travel to food from their home cage to two very differently located sites (chosen according to the rules mentioned in the previous section). The two locations were trained sequentially so that one site was trained first (target A) followed by training on the second site (target B). A probe trial without food after training target A was designed to check if mice had learned it successfully. Only one hole was filled with food in any given training trial. In a second probe trial after training on B, mice were tested to see whether they can travel between the two remembered locations (food is absent; probe B-A trial) despite no prior training nor reinforced experience on the short cut route (*Figure 1E*).

Mice successfully learned the location of target A, evidenced by their direct search trajectories and significant decreases in distance traveled (*Figure 6A*), with the same performance as in the static entrance case. During the learning of target A, the distance of the mouse to the future location of target B was always larger than that expected by chance, that is to a randomly chosen location in the maze (see Materials and methods, *Figure 6—figure supplement 2*). In fact, the trajectories stayed, on average, approximately 40 cm away from the forthcoming target B position. In other words, the mice did not learn, by chance, direct unreinforced routes from 'near B to A'. In *Figure 6A*, for example, the mouse passed within 16 cm of the future target B site in an early learning trial, but this close approach does not appear to constitute a B->A trajectory. The first probe trial (without food) confirmed that mice had learned the location of target A. After not finding food in A, the mice re-initiated a random search (*Figure 6B*).

After the probe trial, mice were trained to learn target B. During this training, three mice (Mouse 33, 35, 36) first learned a route to target B but, on subsequent searches, they would sometimes first check the site of the previous target A and then travel to the correct target B along a direct short cut route (see Discussion). Nevertheless, on average the mice kept farther from the previous site A than the expected by chance (*Figure 6—figure supplement 2*). A first visit to B then to A was never observed during training because mice go back home after getting the food in B.

After learning the location of target B (*Figure 6C*), food was also omitted from this target for Probe B-A. As illustrated in *Figure 6D*

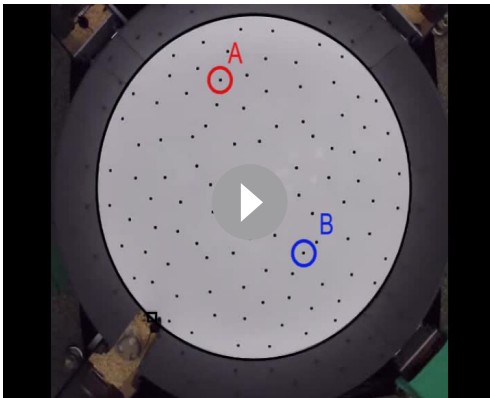

**Video 1.** The Shortcut video illustrates a two-food site experiment including hole checks; there were no landmarks. The mouse was first trained with food at Site A (6 days, 18 trials) and, after training was complete, trained with food at Site B (3 days, 9 trials). The video was taken on a probe trial (Day 10) with no food at either Site A or Site B. The mouse is observed to proceed along a TEV to Site B with a departure from the TEV at the second hole check, followed by a third hole check as it loops back to Site B. After hole checking Site B, the mouse goes on a TEV to Site A with a departure after its third hole check. It then returns to the TEV with further hole checks. Just before reaching Site A the mouse turns and moves very slowly to a hole 16 cm from Site A; this hole check was missed by our algorithm and had to be manually added for statistics. The mouse spends ~2 s at this hole, although it had never contained food; other holes were visited for <1 s. Over the course of the experiment, the mouse visited this hole five times prior to finding the food, but did not visit it in the four trials preceding this Probe trial. Other holes within 15 cm of the target were visited at equal or greater rates. In other words, there was no special sensory input that would have made this hole interesting. We hypothesize that this was the hole predicted by the mouse's cognitive map to be Site A. The mouse subsequently continued to check Site A and a site near the wall. At this point the experiment was terminated. Our 'wash and rotate the floor' protocol to eliminate odor trails or 'floor scratch cues' results in the floor of the maze being washed and rotated 108 times from the first Site A learning trial to the final Probe trial when Site B and Site A were empty. The floor rotation results in the same configuration of hole positions relative to every entrance, making the environment identical at every trial for all the mice (see Materials and methods). Note that the mouse makes its first hole check at a hole near the entrance and a final hole check near the maze wall far from an entrance. Food was never given at either site and there were no features differentiating these holes to make them 'interesting'. These 'near the wall' hole checks were also seen in other mice, but we have no compelling hypothesis as to why they occur at these holes.
https://elifesciences.org/articles/95764/figures#video1

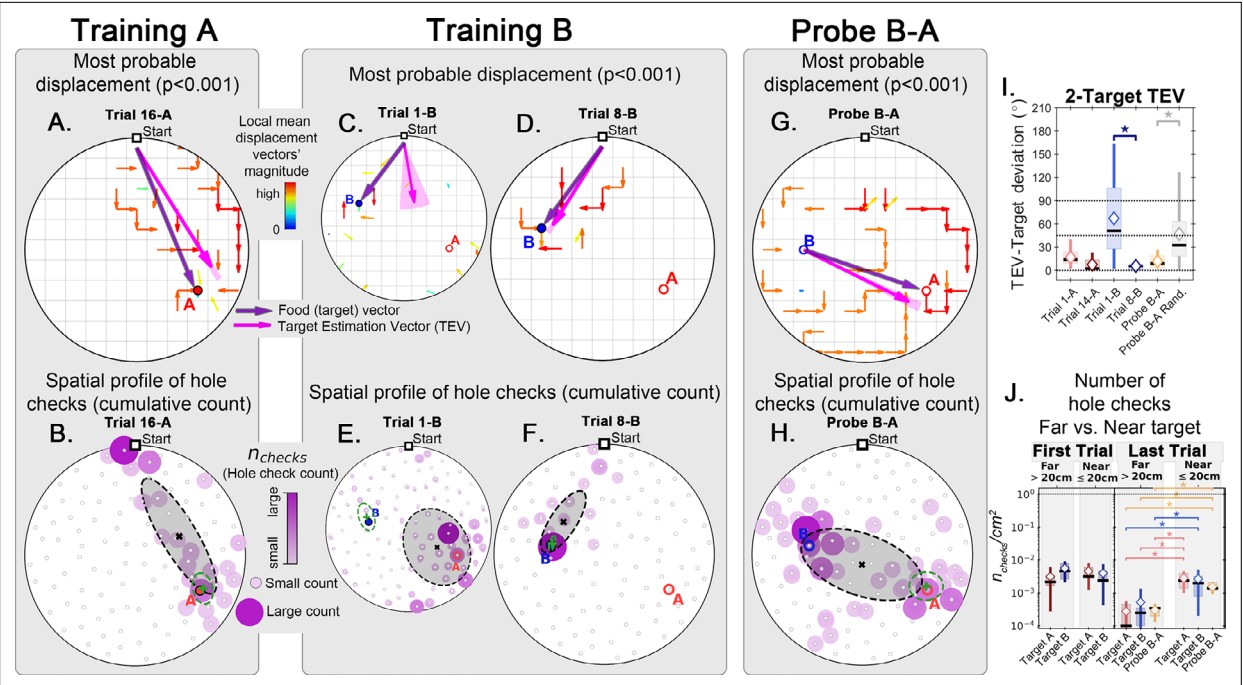

**Figure 7.** Trajectory directionality and active sensing in two food location experiment. Arenas on the top row (mean displacement vector) correspond to the ones immediately below them (hole checking spatial distribution); the red 'A' and blue 'B' labels mark the targets (food sites), which are pointed by the target vector (purple arrow). Top row (**A, C, D, G**): the color and arrows indicate the most probable route taken (red = more probable; only p<0.001 displacements shown; pink arrow = inferred target position, or TEV; shaded pink sector = S.D. of TEV; see Materials and methods, and *Figure 6—figure supplement 1*). Bottom row (**B, E, F, H**): spatial distribution of hole-checks; size and color of circles = normalized frequency at which a hole was checked (larger pink circles = higher frequency); Black ellipse (x=mean): covariance of spatial distribution. Green ellipse (+=mean): covariance of spatial distribution restricted to ≤20 cm of the target. Three stages of the experiment are shown (N=8; all training done in *static entrance with no landmarks*): after learning the target A (**A, B** trial 16 A; significant routes and hole checks are observed only along the target vector, as expected); training of the target B (**C, E**: trial 1-B; **D, F**: trial 8-B); it shows the evolution of the TEV from pointing to A to pointing to B, and the hole checks distribution becomes limited to the newly learned target vector towards B; probe B-A (**G, H**) shows significant routes from B to A (shortcuts; N=5 out of 8 performed the route Start→B→A; see *Figure 2—figure supplement 1* for all samples); hole-checks accumulated along the B-A path suggesting that mice remember both locations. (**I, J**) TEV-target deviation and hole-check area density, respectively. Probe B-A measures are compatible with trials where trajectories have already been learned. Standard boxplot statistics. Diamond: mean. Asterisks/star: p<0.05 (t-test).

and the Shortcut video (*Video 1*) (all mice shown in *Figure 6—figure supplement 2*), five of eight mice were observed to go from B to A via varying trajectories (namely, mice numbered 33, 35, 36, 57, and 59). Two (number 58 and 60) of the remaining three mice went first to A and subsequently to B via direct or indirect routes; these mice had previously taken direct routes to B, suggesting that going to A first is not due to a learning deficiency. The remaining one, mouse 34, went from B to the start location and then, to A. This mouse had previously taken the B-A route during training. In all cases, the mice clearly remembered the location of both targets, even though target A had not been presented or rewarded for 4 days.

The geometric and kinetic features of these experiments for early and late learning of each target, and for Probe B-A, are presented in *Figure 6E–H*. For defining the trajectory quantities in the Probe B-A trial, the 'start' position is taken as target B, and the 'target' is the A site (e.g. the normalized trajectory length is the ratio between total traveled and direct B-A distances). The quantities in probe B-A are statistically indistinguishable from late learning of targets (i.e. trials 14 A and 8-B). Conversely, all the Probe B-A values are significantly different both from trial 1-B and from randomized B-A trajectories (see Materials and methods; p<0.05; except for the density of hole checks which is statistically equal for all considered trials, *Figure 6H*). Therefore, the behavior of the mice when going from B to A in the probe is compatible with the behavior of an animal that acquired spatial memory about the trajectory, even though this trajectory was never reinforced. This suggests that the mouse computed the shortcuts without prior experience (see Shortcut video *Video 1*) for an example of shortcutting and associated hole checks.

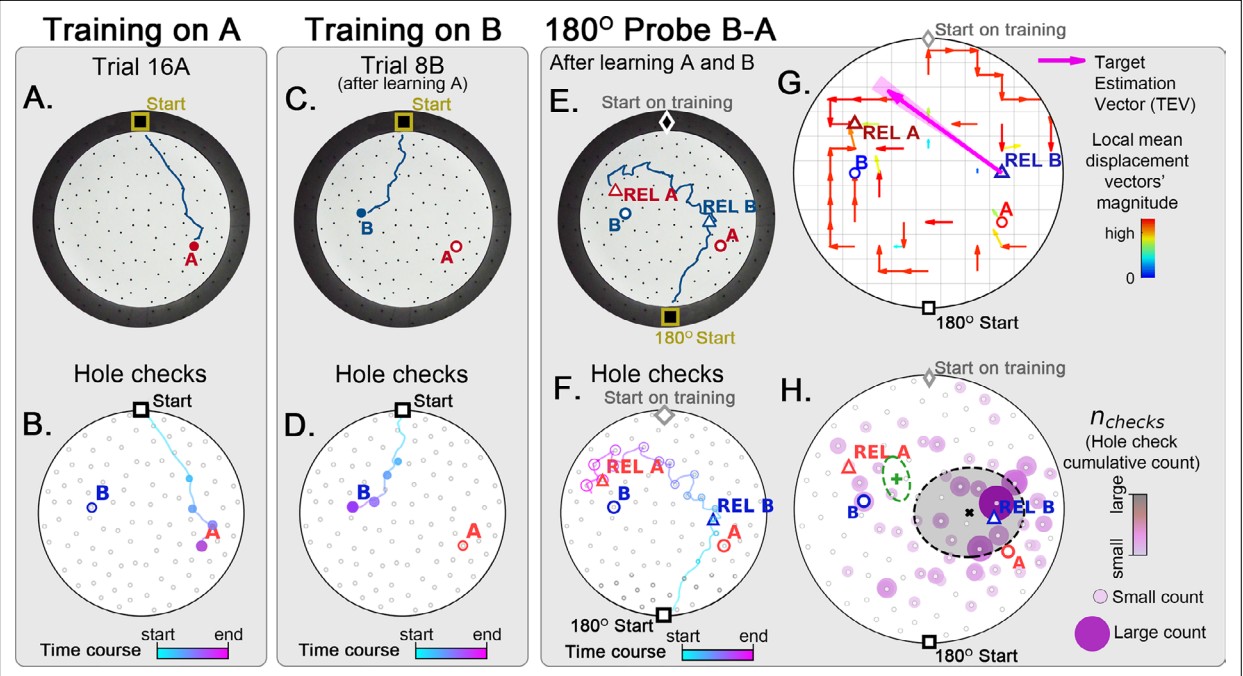

**Figure 8.** Two food location training with 180° rotated probe. Mice (N=8) are trained to find food in the target labeled 'A' (red circle, panel **A**), and in target labeled 'B' (blue circle, panel **B**) afterwards; then a 180° rotated probe trial (no food; panel **C**) is realized to check whether the mice are able to generalize and take the shortcut from REL-B (blue triangle) to REL-A (red triangle), instead of the A and B targets. No landmarks are present. Panels **A, C, E** show exemplars of trajectories in three stages of the experiment, and **B, D, F** show their time course and hole-check locations marked with circles that increase with elapsed time. (**G**) Trajectory directionality analysis and TEV (pink arrow; shaded sector: S.D.) show that significant paths (p<0.001; N=8; see Materials and methods) point from REL-B to REL-A in the same way that it pointed from B to A without rotated entrance in *Figure 7G*. (**H**) The spatial density shows that hole checks accumulate along the REL-B to REL-A direction, instead of the B-A direction in the case without rotation in *Figure 7H*. Black ellipse (x=mean): covariance of hole check density. Green ellipse (+=mean): covariance of the same data restricted to ≤20 cm of the REL-A location. This suggests that mice follow shortcut trajectories anchored to their start location (idiothetic frame of reference).

We calculated the displacement and hole check maps and the TEV for the whole training protocol, including the probe B-A (*Figure 7*). Again, the data show that, after training for A, the TEV pointed directly to A and hole checks accumulated along its path (*Figure 7A and B*). The beginning of training for B (trial 1-B) generated random search patterns, while the TEV and hole checks tended towards A. After learning B, we see that the TEV and hole checks completely shifted to it (*Figure 7C, D, E and F*). Remarkably, the probe B-A trajectories revealed a strong directed flow with a TEV pointing from B to A, whilst hole checks visibly accumulated along this route (*Figure 7G and H*). The TEV-target deviation remained close to zero in late learning and probe B-A (*Figure 7I*), whereas the area density of hole checks is increased near the target compared to far only after learning and in probe B-A (*Figure 7J*). These maps suggest the emergence of a cognitive map guiding the mice when taking the B to A unrehearsed shortcut routes.

As a stringent control, we performed a two-food-locations experiment with a 180°-rotated probe B-A trial (*Figure 8*; N=8). Landmarks were never present (neither for training nor for probe). In the rotated probe B-A, the mice went to the REL of B and then to the REL of A. The displacement map, hole check spatial distribution and TEV now all pointed from REL B to REL A (instead of B to A). This is consistent with both the independent experiments of the rotated probe with static entrance training and of the two-food location. Mice can thus take a novel short cut and have therefore computed a cognitive map based on a fixed starting point and self-motion cues alone.

## Discussion

We have shown that mice can learn the location of a hidden food site when their entrance to an open maze remains the same across trials. Trajectories initially appeared random and took on average, over 100 s and covered a distance of about 10-fold that of the direct route between start and food sites

(*Figure 3*). After a maximum of 14 trials, mice reached asymptotic performance taking, on average, <10 s to reach the food site and with trajectories reaching a near optimal distance: 1.4 times greater than the direct route (*Figure 3F*). In the spatial learning trials, mice frequently checked holes for food along their start to food trajectories. Analyses of hole checks demonstrated the distribution of hole check sites was also modified during learning (*Figure 4G and H*). Hole check density decreased with learning in the first half of trajectories but remained constant during the second half. At higher spatial resolution, hole checks occurred most frequently closer to the food site after learning (*Figure 4K*). Finally, the number of holes sampled near the food site became more consistent with learning (lower entropy: see *Figure 6—figure supplement 1*). We conclude that, given a static entrance, the mice can learn an accurate estimate of food location that guides their trajectories and associated hole check distribution.

Similar results have been reported for rats trained to find food in a virtual reality (VR) 2D spatial navigation task (*Cushman et al., 2013*). During spatial learning, the reward check rate increased as the rat approached the reward edge for both visual and auditory reward location cues. Navigation (trajectories) and reward checking were, however, in register only for distal visual cues. In our experiments, hole checking at a reward site was not dependent on distal visual cues (see REL discussion below) suggesting that a different source (or sources) of spatial sensory input permits registration of trajectories and hole checking during real world (RW) spatial learning.

## What cues are required for learning the food location?

A number of exogenous (tactile, localized and global visual, odorant) and self-motion (optic flow, proprioceptive and vestibular) cues might contribute to the spatial learning we observed. For any cue to be effective, it must be stable under the experimental perturbations we impose, and therefore provide unambiguous information about the location of the food reward relative to the mouse's starting point. Below we describe which cues might be relevant for spatial learning.

### Allothetic cues

#### Tactile

There were no obvious scratches on the maze floor, but we cannot exclude fine scratches detectable by the mice. We wash the floor and randomly turn it between trials so that any scratches or odor trails are not consistently correlated with the food location. We also performed a specific control experiment with an exhaust fan to eliminate food scent. We conclude that tactile cues are likely not used for spatial learning under our experimental conditions.

#### Localized visual landmarks

Rodent vision is important for tasks such as navigating complex environments, finding shelter, prey capture and predator avoidance (see *Saleem and Busse, 2023* for a review). Previous behavioral and physiological studies suggested that our wall cues could be resolved by the mouse visual system (*Jacobs et al., 2009*; *Long et al., 2015*; *Prusky and Douglas, 2004*; *Prusky et al., 2000*; *Saleem et al., 2018*). It was therefore surprising that the mice were unable to find the food site when its start location was randomly switched between trials. A comparison of static versus random start sites after learning revealed that, unlike the static case, there was no reduction of trajectory length or increase of hole checks near the food site in the random start experiments. We considered whether the mice were learning slowly and simply needed more trials to learn the random start site task. We therefore checked if there was any improvement in target estimation over four successive trials. As shown in *Figure 2—figure supplement 1*, actual hole checks over the four trials are no different from randomized hole checks suggesting that the mice are randomly choosing holes to check.

The random start task requires the mouse to learn the relative location of food with respect to a changing view of the landmarks. Rats have been shown to be able to solve this task in the Morris Water Maze (MWM; *Morris, 1981*). Previous studies showed that mice can use distal cues for spatial learning under different experimental conditions, such as in a one-dimensional T-maze, linear track, or when there are multiple highly salient cues (*Morris, 1981*; *Chapillon and Roullet, 1996*; *Hébert et al., 2017*; *Rogers et al., 2017*; *Youngstrom and Strowbridge, 2012*). Our random start task would therefore seem to have all the requirements for spatial learning. The simplest explanation is that, unlike some of the cited papers, the cues we used were not sufficiently salient.

An alternate factor might be the lack of reliability of distal spatial cues in predicting the food location. The mice, during pretraining trials, learned to find multiple food locations without landmarks. In the random trials, the continuous change of relative landmark location may lead the mice to not identifying them as 'stable landmarks'. This view is supported by behavioral experiments that showed the importance of landmark stability for spatial learning (*Biegler and Morris, 1996*; *Biegler and Morris, 1993*) and that place cells are not controlled by 'unreliable landmarks' (*Jeffery, 1998*; *Knierim et al., 1995*; *Save et al., 2000*; *Zhang et al., 2014*). Control experiments without landmarks (*Figure 4—figure supplement 5A and B*) or in the dark (*Figure 4—figure supplement 5C–F*) confirmed that the mice did not need landmarks for spatial learning of the food location.

## Arena geometry

The mice might use the distance and bearing from its start location to the three other maze entrances. Triangulation might then enable it to estimate the food location based on such global features and continuously update its position by using the relative location of the entrances. Our control experiments (*Figure 4—figure supplement 5*) showed that such information is not needed for spatial learning but do not rule out that this information is an additional cue available to the mice under more natural foraging conditions.

## Odor cues

The mouse's cage will be saturated by its own odor. Transferring the mouse to a different cage might enable it to recognize the change in its starting point. However, our maze was designed to eliminate this 'self-cage' odor cue as each mouse stays in its own home cage during the entire experiment. In the Random trials, the entire cage + mouse is moved to a different entrance on each trial. In the Static trials, the cage +mouse is only moved to a different entrance on the REL trials.

Odor cues might come from odor trails left during a preceding trajectory to food, from the odor of the hidden food or from an odor gradient emerging from the mouse's home as it leaves to find food. The odor trails were reduced by washing the floor between trials. Any remaining odour was made unreliable as the floor was rotated between trials. A subfloor beneath the maze floor was covered with crumbled fragments of the same food within the capped hole. We assume that the entire maze was saturated with the food odor and that this would mask the odor coming from the accessible food. Mice would search in holes adjacent to the food without successfully locating the food, suggesting no detectable odour at that distance. In addition, the mouse ran to and searched at the food hole on probe trials confirming that it did not require a food odor to find the food hole.

Mice can also use localized odor sources as landmarks for spatial learning (*Fischler-Ruiz et al., 2021*), and might therefore use odor gradients emanating from their home cage as a cue. We reduced this potential cue by applying negative pressure via exhaust fans covering the cage top that were turned on 5 min before an experiment commenced and its full air volume evacuated 30 times/minute (150 evacuations) to equilibrate it to the room air. After the door to the maze was opened, the fans induced negative pressure to draw air from the maze into the cage and thereby reduce a potential outward gradient (see Materials and methods). The mice were still able to learn the food location after such reduction of olfactory cues (*Figure 4—figure supplement 5E and F*) suggesting that odor gradients are not required for spatial learning under our experimental conditions.

## Idiothetic cues

### Optic flow cues

Although the mice can take varying trajectories, the TEV suggests that they continuously update their knowledge of the direction from start to food throughout their trajectories. This further suggests that the neurons coding for head direction are essential for spatial learning in our maze. Heading direction can be derived from optic flow signals (*Burlingham and Heeger, 2020*; *Horrocks et al., 2023*). In the static entrance experiments, the heading direction is constantly updated throughout the entire run (*Figure 4—figure supplement 3G*), having the mouse veer toward the food only in the last segment of the trajectory (*Figure 4—figure supplement 3Giii*). Hippocampal neurons of rats randomly foraging in a real world or virtual reality environment can develop directional responses based on rich and stable optic flow signals and without requiring vestibular input (*Acharya et al., 2016*). In our maze, landmarks and home cage openings may drive optic flow signaling irrespective of

whether they act as local visual cues. This is consistent with the hypothesis that visual input can guide navigation via a central retinal stream for landmarks and a peripheral retinal stream for optic flow input (*Saleem, 2020*). Our in the dark experiments suggest that other sensory input might also drive the head direction network.

## Vestibular and proprioceptive cues

The vestibular system encodes natural motion (*Cullen, 2019*; *Mohammadi et al., 2024*) and contributes to the generation of head direction responses (*Cullen and Taube, 2017*; *Taube, 2007*). We hypothesize that, in the 'dark' experiments, vestibular cues are a major signal supporting spatial learning. In agreement with VR studies (*Mao et al., 2020*; *Yang et al., 2024*), we hypothesize that contextual visual cues of the starting point and arena boundary anchor directional information derived from optic flow, vestibular and perhaps proprioceptive signals to fixed environmental features.

## A fixed start location and self-motion cues are required for spatial learning

We also considered that mice, unlike rats (*Morris, 1981*), may be unable to learn the random entrance task because it requires associating four different spatial cue configurations with a food location. This would hold when the visual cues were local. In the static entrance task, the mice might be learning a unique configuration of the cues or even the relative location of a single cue and the food location (*Collett et al., 1986*). In the REL experiments we therefore first fully trained the mice with a static entrance and, in a probe trial, randomly switched them to another entrance rotated by 90°, 180°, or –90°. Unlike rats in the MWM (*Morris, 1981*), the mice ran to the REL location (*Figure 5*) clearly demonstrating that they assume that their start location has not changed and that the local visual cues did not guide the mice. We hypothesize that the optic flow signals emanating from the distinct landmarks, derived from low-resolution peripheral retina (*Saleem, 2020*), are invariant to and thus cannot differentiate between the mouse's start sites. Additional control experiments made without landmarks or in darkness also showed that visual input is not required for spatial learning in our maze (*Figure 4—figure supplement 5*). We hypothesize that the minimal conditions for the mice learning the heading direction from start to food is based on self-motion cues (optic flow, proprioception, and vestibular) and one fixed local cue – the start site.

## Combining trajectory direction and hole check locations yields a target estimation vector

Here we follow a review by *Knierim and Hamilton, 2011* that hypothesized independent mechanisms for extraction of target direction versus target distance information. Our data strongly supports their hypothesis. Target direction is nearly perfectly estimated at trial 6 (*Figure 4I* and Results). The deviation of the TEV from the start to food vector is rapidly reduced to its minimal value (5.16°) and with minimal variability (SD=0.20°). Learning the distance from start to food is also evident at trial 6 but only reaches an asymptotic near optimal value at trial 14 (*Figure 3F*). The learning dynamics are therefore very different for target direction versus target distance. As noted below, the food direction is likely estimated from the activity of head direction cells. The neural mechanisms by which distance from start to food is estimated are not known (but see *Engelmann et al., 2021*).

Averaging across trajectories gave a mean displacement direction, an estimate of the average heading direction as the mouse ran from start to food. The heading direction must be continuously updated as the mice runs towards the food (which is suggested in *Figure 4—figure supplement 3G*), given that the mean displacement direction remains straight despite the variation across individual trajectories. Heading direction might be extracted from optic flow and/or vestibular system and be encoded by head direction cells. However, the distance from home to food is not encoded by head direction signals.

The mice, after learning with static entrances, made the greatest number of hole checks in the vicinity of the food-containing hole. We therefore used the mean of the 'near food' hole check distribution (see *Figure 4*) to give a magnitude to the displacement direction, thus generating the TEV. With learning, the TEV rapidly converged to closely align with the direct start-to-food vector (*Figure 4E, F, and I*). In REL trials, the calculated TEV pointed to the REL despite the lack of food at the hole check sites (*Figure 5*). A close analysis of individual trajectories revealed that, while some closely aligned to

the TEV and went directly to food, others deviated from the TEV/direct route and then returned to it (*Figure 4F* and *Figure 6—figure supplement 1*). The standard assumption is that deviations from a direct route are due to path integration errors (*Etienne et al., 1996*). It is not obvious why errors only occur on some routes. A second possibility is that the deviations are intended and meant to prevent route predictability and therefore predation (*Jun et al., 2014*). It is not clear why the mouse does hole checks if it is only reducing route predictability. A plausible hypothesis is that the mice deliberately deviate from the TEV in order to continue exploring for food-containing holes, *en route* to exploit the food reward. We hypothesize, following *Knierim and Hamilton, 2011*, that a path integration mechanism operates continuously to return the mouse to its current TEV estimate no matter what the reason for the deviations from the TEV.

We hypothesize that path integration over trajectories is used to estimate the distance from start to food. The stimuli used for integration might include proprioception or acceleration (vestibular) signals as neither depends on visual input. Our conclusion is in accord with a literature survey that concluded that the distance of a target from a start location was based on path integration and separate from the coding of target heading direction (*Knierim and Hamilton, 2011*). Our 'in the dark' experiments reveal the minimal stimuli required for spatial learning – an anchoring starting point and directional information based on vestibular and perhaps proprioceptive signals. This view is in accord with recent studies using VR (*Mao et al., 2020*; *Yang et al., 2024*).

Path integration uses self-motion signals to update the animal's estimated location on its internal cognitive map. Path integration gain has been shown to be plastic and regulated by landmarks (*Jayakumar et al., 2019*). Remarkably, a recent study has revealed that path integration gain can also be directly recalibrated by self-motion signals (*Madhav et al., 2024*), albeit not as effectively as by landmarks (*Jayakumar et al., 2019*; *Madhav et al., 2024*). An interesting question for future research is whether self-motion signals can also recalibrate the coordinates of a cognitive map. From this perspective, the Target B to Target A shortcut requires transformation of the cognitive map coordinates so that the start point is now Target B.

Extensive research has shown that external cues can control hippocampal neuron place fields (*Chen et al., 2013*; *Knierim and Hamilton, 2011*; *Muller and Kubie, 1987*) and the gain of the path integrator (*Jayakumar et al., 2019*), making the failure of mice in our study to use such cues puzzling. The failure to use landmarks may be related to our task being low stakes and our pretraining procedure teaching the mouse that such cues are not necessary. Our results may not generalize to more natural conditions where many reliable prominent cues are available, and where there is urgency to find food or water while avoiding predation (*Lai et al., 2024*). Under these more naturalistic conditions, the use of distal cues to rapidly find a food reward is more likely to be observed.

## Implications for theories of hippocampal representations of spatial maps

The TEV is learned using self-motion cues alone and we hypothesize that it guides locomotion to the food hole via a path integration mechanism. This conclusion is in accord with an extensive literature that emphasizes the importance of self-motion cues and path integration for spatial learning (*McNaughton et al., 1996*; *Etienne and Jeffery, 2004*). The conclusion is also not surprising given studies on weakly electric fish that demonstrate that, with a fixed initial location, active sensing and self-motion cues are sufficient for learning the location of a food site in the dark (*Engelmann et al., 2021*; *Jun et al., 2016*), and that accumulation of path integration error degrades performance as a function of trajectory length (*Mirmiran et al., 2022*; *Wallach et al., 2018*). Given the presumed accumulation of error by the mammalian path integration mechanism (*Etienne et al., 1996*), it is generally assumed that proximal and/or distal exogenous cues must calibrate the putative path integrator in order to determine not only target direction but also target distance from a start site (*McNaughton et al., 1996*; *Knierim and Hamilton, 2011*; *Jayakumar et al., 2019*; *Etienne and Jeffery, 2004*). Path integration gain of hippocampal neurons is a plastic variable that can be altered by conflicts between self-motion cues and cues and feedback from landmarks (*Jayakumar et al., 2019*). The independence of the TEV from landmark cues (REL experiments) again demonstrates that, in absence of such 'conflicts', self-motion provides consistent cues for path integration and spatial map formation.

The use of hole checking to compute a mouse's estimate of the food location allows us to refine these conclusions. The TEV provides not only an estimate of the food location, but also of the distance

from start to food site; this is especially clear in the REL experiments where, given the lack of food, a mouse's search is clearly centered at the expected food location (*Figure 5*). Sophisticated analyses have been used to link the spatial coding neurons of entorhinal, subicular and hippocampal neurons to behavioral studies on the interaction of exogenous and self-motion signals (*McNaughton et al., 2006*; *Savelli and Knierim, 2019*). Here, we provide strong behavioral evidence to support the role of hippocampal place cells in encoding the trajectories and food site locations observed in our study. Two studies suggest that, in the rat, CA1 place fields will remain stable in the absence of visual cues. Many place cells responses observed in the presence of visual cues will remain after these cues are removed; the authors conclude that self-motion cues are sufficient to maintain normal place fields (*Chen et al., 2013*). Experiments with blind rats have shown vision is not necessary for the development of normal firing of hippocampal place cells (*Save et al., 1998*). Together, these studies suggest that mice CA1 cells will exhibit place fields in our open maze. Analyses of spatial learning in VR versus RW (rats) suggested that distal visual cues, and self-motion (i.e. proprioceptive, vestibular) cues may be required to activate place cells representing allocentric space. In the absence of distal visual cues, CA1 cells preferentially encoded distance traveled during learned trajectories toward a food goal (*Aghajan et al., 2015*; *Ravassard et al., 2013*).

Rats can learn to navigate, in VR, from different start locations to a hidden goal using very salient distal visual cues (*Moore et al., 2021*). Unlike RW spatial learning, CA1 pyramidal cells then only weakly encoded allocentric spatial information (place fields) but instead, primarily encoded trajectory distance and head direction – thus, these cells may comprise a possible neural implementation of the behavioral TEV. These experiments demonstrate great flexibility in the hippocampal encoding of trajectories and location both across pyramidal cells and, for individual cells, across the learned trajectory. An important question is how CA1 pyramidal cells will discharge as a mouse runs along the stereotyped trajectories learned with only self-motion cues as in our experiments. A stringent prediction of the discussion above is that mouse CA1 cells activated along trajectories towards the hidden food should be activated at equivalent locations in REL experiments, and in response to the same multiplexed cues: trajectory distance, head direction and allocentric location.

A subset of hippocampal (CA1) neurons in the bat were reported to encode the direction and/or distance of a hidden goal (*Sarel et al., 2017*). The vectorial representation of goals by such cells could be the substrate of the start-to-food location trajectories we have observed. Recently described CA1 convergence sink (ConSink) place cells (*Ormond and O'Keefe, 2022*) may also have the properties needed to account for the learned start-to-food trajectories we observed. ConSink cells are directional place cells that can encode local direction towards a goal. ConSink cells will, with training, shift their direction tuning to a new goal. The ConSink population vector average, like the TEV, then points directly from start to goal. In both cited studies, landmarks were present, and it is unknown whether such cells will be found in the absence of visual cues.

ConSink-like cells were reported in the CA1 of a mouse foraging in an open field, but their direction tuning was not pointed towards a singular goal (*Jercog et al., 2019*). We hypothesize that in the pretraining foraging phase of our spatial learning task, ConSink cells will also have random direction tuning. Upon fixed start location training, we hypothesize that the ConSink direction tuning will become aligned with the mouse's trajectories and their population average will closely approximate the computed behavioral TEV. The TEV and the ConSink cell population average are statistical measures derived from behavioral and electrophysiological data respectively. An important question is whether an explicit TEV is computed and defines the spatial map guiding the mouse's food-finding trajectories. An equally plausible hypothesis is that the 'spatial map' remains a distributed computation in the CA1 targeted neural networks. Experimental tests of these alternatives address an essential question: how is spatial information represented in neural networks?

Numerous studies have reported that goal sites are overrepresented by CA1 place cells (*Nyberg et al., 2022*). The requirements for such overrepresentation are that there are stereotyped trajectories directed towards an invisible memory-based goal associated with reward (*Nyberg et al., 2022*). The stereotyped trajectories and the hidden memory-based goal of our study imply that these requirements are met. Interestingly, the 'goal-related place cells' are activated before the goal is reached (*Nyberg et al., 2022*), just as hole checks mostly occur as the mice approach the food containing hole from any direction (*Figure 4F and H*). We hypothesize that, after learning, CA1 place cells will

overrepresent the maze region containing the goal location, and that their place fields therefore overlap the hole check sites surrounding the hidden food hole.

Behavioral time scale plasticity (BTSP) has been proposed to be the cellular mechanism that generates new CA1 place fields at important locations, including those associated with reward (*Bittner et al., 2015*; *Bittner et al., 2017*; *Milstein et al., 2021*). BTSP operates up to a ~3 s time frame. If BTSP is operating during spatial learning in our maze, there will be excessive place fields within the <3 s search time before it finds the food hole. BTSP is bidirectional and can result in CA1 place fields translocating with experience (*Milstein et al., 2021*) and the location of CA1 cell place fields may therefore evolve during static entrance training. We note that the cited experiments were done with virtual movement constrained to 1D and in the presence of landmarks. It remains to be shown whether similar results obtain in our unconstrained 2D maze and with only self-motion cues available.

The putative emergent CA1 place fields might be randomly distributed but might also be connected with the 'special' hole check locations. We plotted the evolution of <3 s from target hole checks in the static and random entrance experiments (*Figure 3—figure supplement 1*). With spatial learning (static entrance), temporally 'close to target' hole checks increase relative to temporally distant hole checks and converge to the target site. Active sensing is critical for electric fish spatial learning (*Engelmann et al., 2021*), and is also known to potentiate or induce CA1 cell place fields (*Monaco et al., 2014*). We hypothesize that the persistent <3 s hole checks near the food site that increase during training will, via the BTSP mechanism, drive the emergence and translocation of CA1 cell place fields so that they accumulate centered on checked holes near the rewarded food site (*Nyberg et al., 2022*). This leads to the prediction that a place cell discharging during a hole check near the food site should also discharge after the mouse start site has been rotated and it checks an empty REL hole.

## Shortcutting – evidence for a cognitive map derived from self-motion signals

The O'Keefe and Nadel text (*O'Keefe and Nadel, 1978*) connected hippocampal place cells to the abstract concept of a cognitive map developed by *Tolman, 1948*, and this linkage has been generally supported with few dissenting views (*Bennett, 1996*; *Grieves and Dudchenko, 2013*; *Benhamou, 1996*; *Shamash et al., 2021*). The criteria for a cognitive map are of animals taking unrehearsed shortcuts (*Tolman et al., 1946b*), detours (*Tolman, 1948*) or novel routes (*Morris, 1981*). In this literature, animals typically have both landmark and self-motion cues available for spatial learning. To the best of our knowledge, unrehearsed shortcut behavior using only self-motion cues and a fixed start location has only been shown in humans (*Etienne and Jeffery, 2004*; *Landau et al., 1984*). Our result on shortcutting after spatial learning based entirely on a fixed start location and self-motion cues (*Figure 7*) is therefore the first behavioral demonstration of a rodent cognitive map learned without exogenous cues and using the strict Tolman definition. The TEV for the shortcut trajectories well approximates the direct route between the Site B and Site A food locations (*Figure 7G and H*) demonstrating the accuracy of the putative cognitive map food location estimate.

The shortcut trajectory from Site B to Site A (*Figure 7G and H*) might be formally computed in two ways. For the three mice that had taken the Site A to Site B route during training, the following vector arithmetic is required for the final probe trial when it went from Site B to Site A:

where $\overrightarrow{TEV}_{B \to A} = -\overrightarrow{TEV}_{A \to B}$. For the 4 mice that had never taken the Site A to Site B route during training, the following vector arithmetic is required for the final Site B to Site A shortcut:

$$\overrightarrow{TEV}_{B \to A} = \overrightarrow{TEV}_{Start \to A} - \overrightarrow{TEV}_{Start \to B}$$

The TEVs for Start→A and Start→B appear to be still remembered by the mice in the probe trial, since the accumulation of hole checks at both sites is still evident (*Figure 7H*). The hypothesized ConSink place cells directed to Targets A and B and the accumulation of cells with place fields at both sites will therefore, as described above, still encode the trajectories to each location. To our knowledge, neurons that might compute the required vector arithmetic have not been identified in any part of the rodent brain.

We hypothesize that the TEVs estimated by neural networks downstream of goal vector cells, CA1 ConSink cells and goal location place field cells will be used to compute the shortcut trajectories.

Discovering the networks that do these putative computation(s) may provide insight into the neural bases of the spatial cognitive map.

## Materials and methods

### Animals

All animals were housed in the University of Ottawa Animal Care and Veterinary Services (ACVS) facility. C57Bl/6 wild-type male and female mice were ordered from Charles River, arriving at 8–9 weeks old. Mice were individually housed in 12 hr light/12 hr dark cycles (lights on at 11:00PM EST, lights off at 11:00AM EST). Animals had had food and water available ad libitum. The temperature of the room was kept at 22.5 °C and the humidity was 40%. Mice were habituated in ACVS facilities for 1 week and began testing when they were 10–11 weeks old. Testing of each cohort took place over the course of approximately 2 weeks, 1 hr after the lights turned off. Subjects weighed 22–27 grams at the start of behavioral training. Both male and female mice were used; the same results were obtained in both sexes and were therefore pooled. All animal procedures were conducted with the approval of the University of Ottawa's Animal Care Committee and in accordance with guidelines set out by the Canadian Council of Animal Care.

### Apparatus design and setup

The Hidden Food Maze (HFM) is a framework that trains mice to search for a food reward hidden in an open, circular arena (*Figure 1*). The protocol for the task was inspired by an open maze protocol used to study electric fish spatial learning (*Jun et al., 2016*) and by the Cheese Board spatial task in which rats are placed inside an arena with many holes in the floor, one of which contains a food reward (*Kesner et al., 1991*). Unlike the Kesner et al task, the HFM does not require the animal to be handled, and the pattern of holes is arbitrary rather than grid-like. This task design was also chosen to mirror the setup of the Morris Water Maze (MWM; *Morris, 1981*), which can test allothetic or idiothetic navigation. Like the MWM and the electric fish arena, mice are searching for a food reward location in a circular environment after starting from one of four starting home locations. External landmarks can, if desired, be placed on the maze walls or local cues placed in the maze. In contrast to the MWM, this spatial learning task is a dry maze that is food motivated, which is more naturalistic and less stressful than motivation by the aversion to swimming and fear of drowning. Notably, the task has specific design features to control for against unwanted cues, such as odours, visual cues, and handling.

The maze has a removable floor that is washed & rotated between trials to eliminate odor trails which mice from previous trials might leave behind. The circular floor is 120 cm in diameter and has 100 holes (1.2 cm diameter) randomly dispersed throughout the surface, with 25 holes in each quadrant. The distance between each hole is, on average, 10 cm. The pattern of the holes is rotationally symmetrical, so the pattern looks the same regardless of whether the floor is rotated 90°, 180°, or 270° with respect to the mouse's entrance. This ensures that the mouse will have the same initial view of the holes regardless of which entrance it starts from and how the floor is rotated. Each hole is encircled by a 1 cm plastic rib, sticking downwards, which can be capped at the bottom to hold food that remains invisible from the surface. Thus, the mice cannot discern the contents of the hole just by looking across the floor from their home, but instead need to approach the hole and look inside. The surface of the floor is sanded to be matte to avoid generating reflections from the lights which might interfere with the cameras or distract the mice.

The maze floor is circular and uniform from the inside so as not to provide any directional cues. The floor of the arena is encircled by tall black walls. The walls are made of solid black PVC plastic which forms a cylinder around the maze and is open at the top. The walls are 1 cm thick and 45 cm tall, which is tall enough so the mice cannot jump out. The walls are symmetrical and designed to eliminate geometric cues that would give away directional information from asymmetries in the environment shape. Mice can use odors as landmarks for spatial learning (*Fischler-Ruiz et al., 2021*) and we wanted to eliminate local food odour from a filled hole as a landmark. The arena is resting on a subfloor that contains crumbled food; food odor will diffuse through the open holes and saturate the maze thus masking the odor from a food-filled hole.

The home cages attach to the main arena by being slotted into each entrance. The home cages are 27cm x 16.5 cm x 45 cm and are open at the top. They contain a food hopper and a water bottle

feeder. The dimensions of the home cage are based on commercial mouse cages and comply with the Canadian Council on Animal Care mouse housing standards.

When moving mice to a different starting quadrant, the home cages are designed to slide out from the maze and into a new entrance so the mice do not need to be handled and will therefore not be stressed. The home cages include their own doors, separate from the doors that provide entry into the maze, so the home cage can be freely moved and the mice remain securely confined. The detachable home cages allow the mice to be moved to different starting locations, allowing us to control against navigational strategies that rely solely on response learning or path-integration from a fixed starting point.

Mice can perform the entirety of the trial without experimenter handling. The maze doors are designed to slide upwards and provide access to the main arena. When the trial is finished, the experimenter can slide the doors back into place and the mouse is confined to its home cage once again. To not disturb the mice, the doors are left open while they navigate.

Four LED flashlights were aimed at a white ceiling in order to create dim, diffuse lighting throughout the maze. The illuminance is measured to be 50 lux at the surface of the arena. Dim light was used to allow the mice to properly see all visual cues as well as the maze details. This procedure is assumed to not perturb the nocturnal cycle of mice (*Kronfeld-Schor et al., 2013*; *Upham and Hafner, 2013*). Black curtains surrounded the maze to prevent the interference of non-controlled visual cues.

We used additional experiments to control for possible visual or odorant cues from the open home door. We did some experiments in total darkness using infrared LEDs with emission spectra detected by our camera. In order to exclude any potential olfactory cues emanating from the open cage door we did experiments where odor absorbent kitty litter was placed on the cage floor and two connected exhaust fans (AC Infinity MULTIFAN S5, Quiet Dual 80 mm USB Fan) placed above the home cage. At their lowest (quiet) setting, the fans drew air from the home cage and blew it to the outside; replacement air from the outside came in via small openings at the bottom of the home cage. The home cage air volume was evacuated 30 times/min, effectively equilibrating the cage and open maze air before the doors were slid upwards. Fans were turned on for 5 min (150 evacuations) before the trial started and for the duration of the trial. After the door was opened, the fans induced negative pressure to eliminate diffusion of odors from the home cage to the maze. The two fans were placed over each home cage and turned on to eliminate a potential directional noise cue.

## Behavioral training

### Pre-training

Five days before training, mice are transferred from standard animal facility cages to the experimental home cages for habituation. On the first day, mice are allowed to habituate to their new home cages. On the second day, the door dividing the home cage and the arena is removed and each mouse is given free access to explore the empty arena for 10 min. There are no extra-maze landmarks present during pretraining. Animals are food restricted, with 10% of their body weight in food given back each day which they could consume ad libitum. On day 3, randomly chosen 50% of the holes in the arena are filled with food treats (a piece of Cheerio), and each mouse is given 10 min to forage for food. This is repeated on day 4, where 25% of the arena's holes are filled with food treats. On day 5, only four holes placed at the maze center contained treats. Mice pass the pre-training stage when they successfully found treats at all locations within 20 min. Mice mostly confined their search to circular trajectories near the wall of the maze for Days 3 and 4 but, after training with food near the maze center (Day 5), they checked holes throughout the maze (*Figure 1—figure supplement 1A*). Training in the spatial learning task then commenced.

### Visual cues + randomized entrances training

The mice are trained to locate a food reward that has a fixed relationship with four visual cues on the walls. The protocol mimics classic Morris Water Maze setups to test allocentric landmark-based learning. One of the holes in the area is capped from the bottom with a food-containing insert that is not visible from the surface. Extra-maze landmarks (described in the Visual Cues section) are placed on the walls of the maze to serve as location cues. At the start of the trial, the door is slid upwards so the mice can enter the maze, and the mice are given a maximum of 20 min to find the target hole. The home door is kept open to permit the mice free access to return to their home cage during the trial;

preliminary experiments indicated that closing the door perturbed the mice. The door is re-inserted after the mouse has found the food reward and returned home. If the mouse has not returned home by itself after 1 min of finishing the food reward, it is gently guided back by the experimenter. There is an inter-trial interval of approximately 20 min.

Initial experiments used two trials per day but this was increased to speed up learning; our analyses and graphs are truncated at 14 trials to permit averaging over all the mice. In most experiments, three trials a day were given, over 6 days; at the end of each trial the mouse's home is moved to a different entrance. The home location was changed for each trial; the floor was washed and rotated by 90°, 180°, or 270° between trials, independent on whether the home entrance was the same (static trials) or rotated (random trials). The location of the entrance is randomized with the following constraints: no two trials in a row have the same entrance, and all entrances are selected the same number of times so there is no bias. On the 7th day, a probe trial is given where the mouse is allowed to search for food in the absence of any food. Learning continues over 1 more day and, on the 9th day, a reversal trial is given where the visual landmarks are rotated by 180°. The location of the home cage is randomized before every trial. In this task, the mouse would have to learn the invariant relation between the food hole and up to four visual cues in order to locate the food.

## Visual cues

Four black symbols on white backgrounds: a square, a cross, vertical bars, and horizontal bars. The cues are taped 5 cm above the floor. The square was 15 by 15 cm. The cross consisted of two 2.5 cm wide and 14 cm long bars. The four vertical and four horizontal bars were 2.5 cm wide and 14 cm long. Previous behavioral studies have shown that mice can discriminate the visual stimuli we use (*Jacobs et al., 2009*; *Prusky and Douglas, 2004*; *Prusky et al., 2000*). Recording of neurons in mouse V1 have additionally shown that orientation selective cells in mice visual cortex can discriminate our visual stimuli (*Long et al., 2015*).

## Visual cues + static entrance training

This protocol allows mice to navigate by potentially using visual cues in cooperation with path integration or by path integration alone. The setup is the same as visual cues training, except mice enter from the same entrance each trial instead of a randomized entrance. Mice are considered well trained once their latency learning curve has plateaued for three successive trials. The mice's latencies had plateaued by 14 trials but training continued till 18 trials.

After 18 trials, reversal trials were done with no food present and the mouse's home cage rotated 180°. The mouse was allowed to search for food for 10 min. If the mice were able to learn the invariant landmark/food spatial relationship, we would expect them to search at the learned food site. If they had used path integration of self-motion cues to navigate from the fixed entrance to the food, we would expect them to search at the rotationally equivalent location (REL).

## Rotationally equivalent location (REL)

The four quadrants of the arena's floor are identical. This means that the position of the holes in the second quadrant (Q2, *Figure 1B*; also *Figure 4—figure supplement 3A*) is equal to the position of the holes in the first quadrant (Q1) rotated by 90°Counter clockwise (CCW) about the center of the arena. Q3 is Q1 rotated by 180° (CCW) and Q4 is Q1 rotated by –90° (i.e. 90°CW). In other words, every hole in Q1 has an REL in each quadrant achieved by the respective rotation. The REL target works in the same way: it is the position (relative to the entrance used in the trial) where the food would have been had the mouse been trained from the entrance it used in the trial. To illustrate this, consider *Figure 1D*: we put the food (target) in a particular hole in Q1 for a mouse that is trained to enter from Q2 (such as the example in *Figure 1B*) in the static entrance protocol. The vector that points from the mouse's start point (in Q2) to the target in Q1 (i.e. the target vector) is 'anchored' to the start position at Q2. For example, in the mouse's perspective, the target vector is equivalent to going forward for 70 cm, and then turn left and follow for another 30 cm. If in a probe trial, we now let the mouse enter from Q4 instead, there are two options: either the mouse uses the landmarks and searches for the food in Q1 (where the food actually used to be), or it follows the target vector (70 cm forward +30 cm left) and goes to the REL location of the target in Q3. Following the target vector is a sign of path integration using self-motion signals (idiothetic cues).

## Control experiments

### Rotations following static entrances – with and without visual cues

We performed additional tests for the use of visual landmarks in the Visual Cues + Static Entrance protocol. Well-trained mice had their home cages rotated to another quadrant and tested on how well they found the food from a new starting location. Mice were rotated 180°, 90°, and then –90° with respect to their original location (*Figure 5A, B and C*). Mice were given six consecutive trials at each new location. One control group of mice had the visual cues on the arena wall removed during rotation training. If mice were able to use visual cues, we would expect them to improve their search efficiency when visual cues were available. Lack of improvement or similar performance compared to the control group without visual cues suggests that the mice do not rely on the type of visual cues we provided for spatial learning and navigation.

### Navigating and training without visual cues and in darkness

To control for the effects of all visual information, one cohort (four mice) was trained and tested without any cues (*Figure 4—figure supplement 5A and B*). A second cohort (four mice) was trained in the light according to the Visual Cues +Static Entrance protocol, then the lights are turned off and the mice are given a trial in darkness (*Figure 4—figure supplement 5C and D*). The lights were opened in between trials so the mice did not acclimatize to the darkness. Mice were tracked using infrared LEDs with emission spectra detected by our camera. The following conditions were tested: darkness during a regular trial with food present in the arena and during a probe trial where there is no food present in the arena; both conditions gave the same results.

We additionally both trained and tested a cohort (four mice) in darkness and with control of potential odors. A recent study has shown that placing sighted mice in darkness impairs entorhinal cortex head direction cell tuning (*Asumbisa et al., 2022*), but here the mice successfully learned to find food and with the same time course of learning (*Figure 4—figure supplement 5E and F*).

### Two food location training

This protocol aims to test flexibility of spatial learning using only path integration (N=8 mice). No visual cues are placed on the arena walls. Mice entered the arena from the same entrance each trial. Food was placed in a target well in 'Target A' and mice were trained to find this location for 18 trials, 3 trials a day. A probe trial ('Probe A') was done after trial 18. For trials 19–26, the food was moved to a different location, 'Target B', and mice are trained to find the new location. We were careful to choose Target B so that Target A and B were not symmetric with respect to the mouse's entrance. A second probe trial ('Probe B-A') was done after trial 26. The purpose of the 'Probe B-A' trial is to check whether mice take a shortcut between B (latest learned target position) and A (first learned target position). For some cohorts, the training continued until trial 34 and a third probe was given.

### Two food location training with rotated probe

The mice (N=8) were trained exactly as in the 'Two food location' experiment described above. However, the mice start location was rotated by 180° prior to the second probe trial. This protocol joins the 'Static entrances with rotated probe' protocol with the 'Two food location protocol', and is designed to provide further support for our path integration hypothesis. Each of the trained targets have their REL counterparts (180° rotated around the center of the arena). Now, the purpose of the rotated 'Probe B-A' trial is to check which of the two options will happen: either mice take shortcuts between B (latest learned target position) and A (first learned target position); or they take shortcuts between REL B and REL A, supporting path integration via self-motion cues.

## Behavioral analysis

Path tracking was done with Ethovision XT15 (Noldus) based on contrast detection. Mice were tracked according to 3 body points at 30 frames per second on a 1080 P USB camera with OV2710 CMOS. Videos were first recorded on AMCap webcam recording software and imported into Ethovision in order to preserve high quality videos. Trajectory plotting was done in Python.

Latency to target was calculated from when the nose point of the mouse enters the arena until the nose of the mouse enters the target hole. Trajectory analyses are based on the nose point sequence

of the mouse. Speed and distance were calculated based on trajectory coordinates exported from Ethovision.

Search bias during probe trials is calculated by totalling the time a mouse spent within a 30x30 cm square centered around the target vs. the RELs in the other 3 quadrants during a 2-min trial (*Figure 2C*, *Figure 3C*).

## Statistical tests

Significance testing was conducted in a manner that was appropriate for each dataset. Paired comparisons utilized the t-test for parametric data and the Wilcoxon signed-rank test for non-parametric data. The significance cut-off was set at 0.05. Statistics were calculated either using R (URL http://www.R-project.org/) or scipy 1.8.0 in python 3.8.2. The statistical significance testing of the trajectory directionality analysis was developed from first principles, since it involves angular variables (the direction of each vector). It is presented in the 'Analysis of trajectories' section ahead.

Figures showing a quantity evolution over trials *Figure 2D–I*; *Figure 3D–I*; *Figure 4I*; *Figure 4—figure supplement 3* have symbols as averages, error bars as standard error, and the shading around the curve corresponds to the full data range (lower and upper shading limits correspond to minimum and maximum sampled data points, respectively).

Boxplot figures *Figure 4J and K*; *Figure 6E–H*; *Figure 7I and J* have the diamond symbol as the average, the thick black line as the median, the box covering the interquartile range (IQR), and the whiskers extend from the box limits to ± 1.5 IQR in both directions. No outliers were detected in any of these plots.

## Randomized trials

For the Two Food Location experiment, we calculated a randomized version of the Probe B-A trial for comparison. It consisted of extracting ten random pieces of the trajectory. Each piece was defined by randomly selecting a pair of points in the trial trajectory that are separated by the B to A distance. Then, the particular quantity of interest was averaged over each piece of the trajectory, and these averages were then averaged to obtain a single value for the 'Probe B-A Rand'. condition in *Figures 6 and 7*. See *Figure 4—figure supplement 3* for a definition of the quantities.

## Statistical analysis of trajectories

### Experiment alignment

The procedure described in this section is applied to calculate trajectory directionality (and hence the target estimation vector, TEV) and the hole checking spatial distributions. Each mouse in the Random Entrance experiment started from a different quadrant of the arena, for each consecutive trial, and the target was fixed relative to the arena (global reference frame). However, in order to increase sampling, we need to coherently align the mice entrances, generating the mouse's perspective reference frame (see *Figure 1—figure supplement 1B–F*). Each experiment batch was performed with four mice, such that at any given trial, a given mouse entered from quadrant 1 (*Figure 1—figure supplement 1C*), another entered from quadrant 2 (*Figure 1—figure supplement 1D*), the next entered from quadrant 3 (*Figure 1—figure supplement 1E*), and the last entered the arena from quadrant 4 (*Figure 1—figure supplement 1F*). Most of the experiments, then, consisted of two different batches of four mice that had to be conveniently aligned to increase sampling. In this figure, we aligned all the entrances to the Start point at the top of the arena (trajectories were rotated accordingly), emphasizing the four possible positions of the target from the mouse's perspective.

We had to find a way to align all the targets in a given trial to one of the four possible positions, such that the experiment stays coherent over trials (i.e. the target randomly switches in between these four positions from trial to trial). For example, in *Figure 1—figure supplement 1B–F*, we show trial 14. The target position for each starting quadrant is labeled with a red letter A and a red circle, whereas the target for the previous trial is labeled with a green circle and a green letter A subscripted with 'trial 13'. These two positions (trial and previous trial) must always be different to keep the random characteristic of the experiment.

Notice that the target positions in each of the panels D, E, and F in *Figure 1—figure supplement 1* are simply rotated relative to the target positions in panel C. The target configuration in panel D is 90°Clockwise rotated relative to the target configuration in panel C. The configuration in panel E is

180°Clockwise rotated, and panel F, 270°Clockwise rotated; both angles are relative to the configuration in panel C. Thus, in order to sample all the mice together, we simply rotate the trajectories from the mice that started in quadrant 2 [panel D] counter-clockwise by 90°; the trajectories from the mice that started in quadrant 3 [panel E] are rotated by 180°Counter-clockwise; and the trajectories starting from quadrant 4 [panel F] are rotated by 270°Counter-clockwise. This reduces all the experiments to the first quadrant, allowing us to sample all the trajectories of all the mice together in each individual trial.

## Active sensing and hole-check detection

We developed an algorithm to detect the mouse's behavior of sniffing holes to detect food as a measure of active sensing. The hole checking events are used to infer the mouse's memory and uncertainty about its environment. The procedure described here is applied independently to each trial. We compute the spatial distribution $P\left(X_i, Y_i\right)$ for the count of hole checks for each hole $i$ positioned in $\left(X_i, Y_i\right)$ in the arena, accumulated across all mice in a given experiment (*Figure 4C, D, G and H*; *Figure 5C*; *Figure 7B, E, F and H*; *Figure 8C*; and *Figure 1—figure supplement 1B–F*; usually N=8 mice for each experiment), and normalized by the total count. The frequency of checks in each hole is coded both in the color and size of the filled circles: larger and darker shaded circles correspond to larger number of checks in that particular hole (lighter shaded pink small circles are a small number of checks). The black ellipsis marks the covariance of the $P\left(X_i, Y_i\right)$ spatial distribution ('x' in the figures marking the mean position of hole checks). It is constructed from the eigenvalues and eigenvectors of the covariance matrix $C$ of the hole check coordinates $\left(X_i, Y_i\right)$ weighted by $P\left(X_i, Y_i\right)$: the eigenvectors give the directions of the ellipsis semi-axes, whereas the eigenvalues give the width of each semi-axes. The center of the ellipsis (mean of the ellipsis' foci) is aligned with the mean of the spatial distribution.

Under the path integration hypothesis, it is expected that error accumulates as the mouse walks (*McNaughton et al., 2006*). This is consistent with the increase in number of hole checks per unit area near the target (*Figure 4J and K*; 'near' = less than 20 cm away from the target). We consider that the change in number of hole checks measures the variability of the mouse's estimate of the target position.

We therefore calculate the mean and covariance of the distribution of hole check events restricted to within 20 cm of the target (again, the green 'x'=mean). The distance from the start to this restricted mean is used to scale the TEV vector (see Target Estimation Vector).

The arena design forces the mouse to put its nose very close to the hole to be able to see the food or detect any food odour. With that in mind, we defined two methods for detecting a hole check (see *Figure 4—figure supplement 1A and B*, and the Shortcut video *Video 1*). The two methods are complementary, and the second can find hole-check events missed by the first method: we apply the first method, then perform a visual check on the data to see if there were potential hole-checks that were missed. Then, we apply the second method to capture any remaining events.

The first method is the minimum velocity criterion and provides a high threshold for hole check identification. Four conditions must be satisfied simultaneously to detect an event: (i) the nose of the mouse must be within 3 cm of an arena hole; (ii) the velocity has to be less than 20% its maximum value; (iii) the velocity must be at a minimum; and (iv) the velocity has to have dropped by at least 5 cm/s to reach that minimum.

The second method is more inclusive and defines a hole-checking event by a simple slowing down event, provided that: the nose is within 3 cm of an arena hole, and the slowing down is enough for the velocity to drop past the 20% threshold of its maximum.

The detected events by the application of these two methods in sequence are marked for all trajectory samples of the Probe B-A in as examples. A particular case is shown in more details in *Figure 4—figure supplement 1*, and in the Shortcut video (*Video 1*) with the recording of a mouse's performance. It is worth mentioning, there is a clear hole-check missed by our algorithm in the 25–27 s range. This is because the velocity of the mouse was already below the 20% maximum before and after the hole check, hence the two sets of criteria defined above were not met for this particular event and it was counted by the manual check. Otherwise, we clearly see that all the other events are captured by sequential use our two methods.

## Trajectory directionality (displacement map)

This analysis is a way to visualize mouse trajectories and directionality across all the mice for each trial. It is related to a velocity map of the arena, except that here, the arrows point in the direction of most probable movement instead of the direction of the velocity. *Figure 4—figure supplement 1B* shows a scheme for how we compute this map, and we detail it below. We call this quantity as the 'displacement map', since it gives the displacement of the mouse for each position in the arena.

In this section, we explain how to calculate the displacement map for a single sample of $N$ mice. The procedure here is applied to each individual trial independently (either during learning, or for a probe trial). This map is used for inferring the learned directionality of the target (relative to entrance). The error, average and significance of this quantity is estimated from first principles by a jackknife procedure explained in the next section. In the figures we show the average displacement map vectors that were found to be significant. These maps are then used to compute the target estimation vector (TEV; detailed in the last section).

We start by overlaying a lattice of size $L$ on top of the arena recording. This means that there are $L$ boxes on the $x$ (horizontal) direction, and $L$ boxes on the $y$ (vertical) direction, making a total of $L^2$ boxes in the lattice. Each box defines a lattice site with coordinates $(x, y)$, such that $x$ and $y$ are integers between 1 and $L$. Each mouse corresponds to an independent observation of a trajectory, so we overlay all mice trajectories for each trial separately in the calculations below. Next, we map the coordinates of the trajectories into the lattice coordinates in order to obtain a temporal sequence of visited lattice sites in each trial. We are interested in counting the number of times that each subsequent pair of adjacent lattice sites appears in this sequence, regardless of what mouse it came from. This will be used to define the displacement map, detailed in what follows.

The displacement map $\vec{M}(x, y)$ is a vector field defined on the lattice that assigns to each site the preferential direction of movement. This direction is estimated from the trajectories data. Mathematically, it can be written as $\vec{M}(x, y) = [M_h(x, y), M_v(x, y)]$, where $M_h$ is the horizontal component and $M_v$ is the vertical component. The horizontal component expresses the trend to move to the left or right (along the lattice $x$-axis), and the vertical component in the perpendicular direction, that is 'up' or 'down' in the top view of the lattice of *Figure 4—figure supplement 1D* (along the lattice $y$-axis). Each component is a function of the lattice coordinates, $(x, y)$. The vector $\vec{M}$ is defined as the spatial gradient of the probabilities to move out of $(x, y)$ towards one of its adjacent sites. Thus, its components are given by

$$M_h(x, y) = P_\rightarrow(x, y) - P_\leftarrow(x, y),$$
$$M_v(x, y) = P_\uparrow(x, y) - P_\downarrow(x, y),$$

(1)

where $P_\rightarrow(x, y)$ is the probability of stepping right, that is from $(x, y)$ to $(x+1, y)$; $P_\leftarrow(x, y)$ is the probability of stepping left [from $(x, y)$ to $(x-1, y)$]; $P_\uparrow(x, y)$ is the probability of stepping up [from $(x, y)$ to $(x, y+1)$]; and $P_\downarrow(x, y)$ is the probability of stepping down [from $(x, y)$ to $(x, y-1)$]. The probabilities of stepping out of $(x, y)$ must be normalized, therefore $P_\rightarrow(x, y) + P_\leftarrow(x, y) + P_\uparrow(x, y) + P_\downarrow(x, y) = 1$. If there is no trajectory going through a particular site, the probability of going from that site into each of the directions is equal to the 'null' probability, $P_0 = 1/4$. Time sequences where the mouse does not move are ignored, since we are only interested in the movement between adjacent sites.

The stepping-out probabilities are computed from the mouse trajectory in the following way. This procedure is applied to all mice overlayed together in each individual trial. First, a step is defined as moving from a box at $(x, y)$ to one of its four adjacent boxes, say the one on the right $(x+1, y)$. Now, if we wanted to calculate the probability of going right from a position $(x, y)$, we need to go through the sequence of visited lattice sites looking for $(x, y)$ in an instant followed by $(x+1, y)$ in the immediately next instant. We count the number of times $S_\rightarrow(x, y)$ that this pair appears. The count of the transitions of $(x, y)$ to $(x+1, y)$, in this example, is made regardless of when these transitions happened during the trajectory. The only condition is that the position $(x, y)$ must be immediately followed by $(x+1, y)$. The same counting is made for the transitions from $(x, y)$ to $(x-1, y)$ and stored in $S_\leftarrow(x, y)$, from $(x, y)$ to $(x, y+1)$ in $S_\uparrow(x, y)$, and from $(x, y)$ to $(x, y-1)$ in $S_\downarrow(x, y)$.

With these sums of steps performed, we can calculate the probability of stepping right, left, up, or down. First, note that initially the chance of going to any direction is $P_0$, assuming we know nothing about the trajectories. Now, given that we observed $S_\rightarrow(x, y)$ steps going to the right, we need to

update the chance of going to the right using the union (i.e. sum) of the observed chance $S_\rightarrow(x,y)/n_s$ with $P_0$, where $n_s$ is the total number of steps counted toward any direction in all lattice sites in a given trial:

$$W_\rightarrow(x,y) = \frac{S_\rightarrow(x,y)}{n_s} + P_0 - \frac{S_\rightarrow(x,y)}{n_s}P_0, \tag{2}$$

where $W_\rightarrow(x,y)$ is the direction weight of the action 'step to the right from $(x,y)$'. Alternatively, this can be written as a weighted sum of memory-driven stepping with probability 1 and random stepping with probability $P_0$:

$$W_\rightarrow(x,y) = \frac{S_\rightarrow(x,y)}{n_s} + P_0\left(1 - \frac{S_\rightarrow(x,y)}{n_s}\right).$$

We employ *Equation 2* because the mouse could, in principle, have chosen any of the other three directions. This ensures that every time the memory is increased (adding $S_\rightarrow/n_s$ to the weight), the random contribution for that step decreases; this justifies the last term in *Equation 2*, where the intersection between the memory term and the 'null' probability, $S_\rightarrow P_0/n_s$, is subtracted from the aforementioned union. One can easily see that the formula guarantees that the null probability is automatically recovered, that is $W_\rightarrow(x,y) = P_o$, when there are zero observed steps in a given direction at position $(x,y)$, and that $W_\rightarrow(x,y) = 1$ if all steps are to the right at position $(x,y)$.

Finally, we use these weights to calculate the probabilities, first defining the total weight of stepping out of $(x,y)$,

$$W_{out}(x,y) = W_\rightarrow(x,y) + W_\leftarrow(x,y) + W_\uparrow(x,y) + W_\downarrow(x,y), \tag{3}$$

and then calculating the probabilities by

$$P_\rightarrow(x,y) = \frac{W_\rightarrow(x,y)}{W_{out}(x,y)}. \tag{4}$$

The same is made for the other directions (left, up and down). *Equation 4* ensures that the probability of stepping out of $(x,y)$ is always normalized. This is then applied to *Equation 1* to obtain the displacement map.

Let us go through the simple example shown in *Figure 4—figure supplement 1C–F*. In this case, the accumulated trajectories in the center box located at $(x,y) = (3,3)$ are such that there are two passes going up and one pass going down. Thus, each direction has the following weight [given by *Equation 2*]: $W_\uparrow(3,3) = \frac{2}{3} + P_0 - \frac{2}{3}P_0 = \frac{3}{4}$, $W_\downarrow(3,3) = \frac{1}{3} + P_0 - \frac{1}{3}P_0 = \frac{1}{2}$, and $W_\rightarrow(3,3) = W_\leftarrow(3,3) = P_0 = \frac{1}{4}$. Applying *Equation 4*, we obtain the probabilities $P_\uparrow(3,3) \approx 0.43$, $P_\downarrow(3,3) \approx 0.29$, and $P_\rightarrow(3,3) = P_\leftarrow(3,3) \approx 0.14$. This results in a step map vector $\overrightarrow{M}(3,3) = [0, 0.14]$. This vector represents the preferred direction of movement out of site $(3,3)$, meaning the mice are likely to pass in the vertical direction, going from bottom to top (hence a positive vertical quantity); and are not picky regarding left or right (hence a null horizontal quantity).

The trajectories of the mice obey box-scaling independently of trial or experimental condition. Box-scaling is a standard method to measure the dimensionality of curves, and is described in *Falconer, 2004*. In other words, the number of boxes needed to cover any trajectory scales linearly with the lattice dimension $L$; see *Figure 4—figure supplement 1F*. This means that the choice of $L$ is somewhat arbitrary, and we chose $L = 11$ to make each box the size of a few centimeters as expected for the size of a place field from a DG cell (*GoodSmith et al., 2017*).

## Trajectory directionality statistical significance

The method above gives a single displacement map $\overrightarrow{M}(x,y)$ and enables the calculation of the TEV $\overrightarrow{D}$ (see below) for each trial of an experiment containing $N$ mice. We have to generate multiple samples for the same experiment using different mice to estimate the significance of the mean displacement direction calculation. We employ a jackknife procedure to achieve this goal. It consists of the 'leave one out' rule: this means that a sample of $N$ mice give $N$ unique jackknife samples, each of which

containing $N - 1$ unique mice. Then, we get $N$ estimates $\vec{M}_k(x, y)$, with $k$ from 1 to $N$, by applying the previously described procedure to each jackknife sample of $N - 1$ mice (instead of the whole sample). Finally, we calculate the average, $\vec{M}(x, y) = \frac{1}{N} \sum_{k=1}^{N} \vec{M}_k(x, y)$ for the displacement map. This is the vector map that we show as small arrows colored from blue to red (*Figure 4A, B, E and F*; *Figure 5C*; *Figure 7A, C, D and G*; *Figure 8C*; and ; we only show statistically significant averages).

The statistical significance of the average $\vec{M}(x, y)$ map in each lattice position $(x, y)$ is built from first principles as follows (see *Figure 4—figure supplement 2*). The main feature of each of the vectors in this map is its direction (i.e. the angle it makes with the positive x-axis). Strong directionality means that the vectors $\vec{M}_k(x, y)$ all pointed roughly in the same direction (i.e. roughly same angles), whereas weak directionality means uniformly distributed angles around the circle (*Figure 4—figure supplement 2A and B*). The stronger the directionality of the sample $\vec{M}_k(x, y)$ the more significant the average $\vec{M}(x, y)$. A simple measure of the spread of the angles (i.e. the directionality strength) is the standard deviation (S.D.) of the sample angles that $\vec{M}_k(x, y)$ make with the positive x-axis: stronger directionality implies a smaller S.D. (*Figure 4—figure supplement 2C*). Thus, the significance ($p$) of directionality is how likely it is for a sample of $N$ uniformly distributed angles to display a given S.D. value collectively.

The S.D. of $N$ angles (with zero average) is $\sigma = \sqrt{\frac{1}{N-1} \sum_{k=1}^{N} \theta_k^2}$, where the $\theta_k$ are uniformly distributed from –180° to 180°. If we could determine the probability density function $\rho(\sigma)$, then the significance would be given by *Hair et al., 2013* $p(S.D.) = \int_0^{S.D.} \rho(\sigma) \, d\sigma$, which is the probability of observing a given S.D. from the sample. In other words, $p(S.D.)$ is the probability that an observed S.D. came from a set of $N$ uniformly distributed angles with zero mean. Thus, given an experimental observation S.D. from the jackknife sample, we will know what is the probability $p$ that the angles from the sample were uniformly distributed in the circle (i.e. have weak directionality). A small $p$ value, therefore, indicates strong directionality (since having small S.D. is very unlikely for uniformly distributed angles – *Figure 4—figure supplement 2D*).

We need to estimate $\rho(\sigma)$ to be able to calculate $p$ for any S.D. Although it has an integral representation similar to the Chi distribution (*Tomé and Oliveira, 2015*), it is easier to make a numerical estimation: we fix, for instance, $N = 8$ (since we have 8 jackknife samples) and generate 10,000 independent $\sigma$ values applying the S.D. formula above to $N$ independent uniform angles $\theta_k$ from –180° to 180°. The normalized histogram of all the observed $\sigma$ is a good estimate of $\rho(\sigma)$ (*Figure 4—figure supplement 2D*). The $p$ value is obtained by numerically integrating $\rho(\sigma)$ from 0 to the observed S.D. value (*Figure 4—figure supplement 2E*).

Now, it suffices to calculate the S.D. of the angles of the vectors $\vec{M}_k(x, y)$ in the jackknife sample (unbiased to make the average vector $\vec{M}(x, y)$ have $\theta = 0°$). An observed $S.D. = 30°$ Corresponds to $p = 10^{-4}$ (i.e. the sample has very strong directionality). We only show in the figures the displacement vectors that have $S.D. \leq 5°$, yielding vanishing $p < 10^{-7}$ (all vectors regardless of significance are shown in for comparison for random and static entrance experiments). Comparing to *Figure 4*, we notice that, as expected, only vectors from trials that are supposed to have random directionality vanish (i.e. trials for random entrance experiments, and trial 1 for static entrance). The vectors in strongly directed flow trials (such as trial 14 in static entrance experiments) remain. The same happens for the two-food location experiment.

## Target Estimation Vector (TEV)

The target estimation vector (TEV) is an estimate of the position of the target based only on the observed trajectories and hole checks of the mice. We write the average TEV as $\vec{D} = \frac{1}{N} \sum_{k=1}^{N} \vec{D}_k$, and $\vec{D}_k = D u_k$ is the TEV for a particular jackknife sample $k$. The magnitude $D$ is inferred from the hole checks and the direction $u_k$ is calculated from the displacement map $\vec{M}_k(x, y)$ as follows. The total sum of the displacement map $\vec{M}_k(x, y)$ over all arena sites $(x, y)$ gives the estimate of the learned direction $u_k = \vec{u}_k / |\vec{u}_k|$, with $\vec{u}_k = \sum_{(x,y)} \vec{M}_k(x, y)$. The magnitude $D$ is the distance from start to the average of the hole check distribution restricted to 20 cm around the target (i.e. $D$ is the distance from the start to the 'x' in the green ellipsis in *Figure 4G and H*; *Figure 5C*; *Figure 7B, F and H*; *Figure 8C*; and *Figure 1—figure supplement 1B–F*). In the case of the Probe B-A trial, we use $D$ as the distance from

B (instead of 'start') to the restricted average of the hole check distribution around target A. In trials where no preferred direction is detected (such as in the random entrances experiment, or in the first trial after switching from target A to B), the magnitude $\left|\vec{u}_k\right|$ is already roughly the distance from start to the arena's center (expressing the randomness in the displacement map), and hence we take the TEV to be just $\vec{D}_k = \vec{u}_k$ instead.

The error in the magnitude $D$ of the TEV is obtained from the covariance matrix $C$ of the spatial distribution of hole checks restricted to less than 20 cm of the target. It is given by $\sigma_D = \sqrt{\lambda_1 + \lambda_2}$, where $\lambda_{1,2}$ are the eigenvalues of $C$. In the plots [**Figures 4, 5, 7 and 8**], we simply represent the covariance matrix by the green ellipses in the same way we do for the uncensored distribution (see Active sensing and hole-check detection). The error in the direction of $\vec{D}$ is simply the S.D. of the angles of $\vec{D}_k$ with respect to the positive x-axis (calculated by shifting $\vec{D}$ to 0°, and making the angles of $\vec{D}_k$ between −180° and 180°, similarly to what is done for the displacement map). This gives the pink shaded circular sector accompanying the TEV in the figures of the displacement maps and the error bars in the TEV-target deviation plot (**Figures 4I and 7I**).

## Acknowledgements

We thank Dr. Érik Harvey-Girard for technical support, and William Moldenhauer de Jesus for joining the short cut experiment video with the hole check data. This work was supported by the Canadian Institutes for Health Research Grant # 153143 to AL and LM and J-C B, a Brockhouse award (493076–2017) to AL and LM, an NSERC award (RGPIN/06204–2014) to AL, an NSERC award to LM (RGPIN/2017-147489-2017) and a grant from the Krembil Foundation to AL, LM and J-C B.

## Additional information

### Funding

| Funder | Grant reference number | Author |
|---|---|---|
| Canadian Institutes of Health Research | RGPIN/04336-2017 | Leonard Maler |
| Canadian Institutes of Health Research | 153143 | André Longtin<br>Leonard Maler<br>Jean-Claude Beique |
| Brockhouse | 493076–2017 | André Longtin<br>Leonard Maler |
| Natural Sciences and Engineering Research Council of Canada | RGPIN/06204–2014 | André Longtin |
| Natural Sciences and Engineering Research Council of Canada | RGPIN/2017-147489-2017 | Leonard Maler |
| Krembil Foundation | | André Longtin<br>Leonard Maler<br>Jean-Claude Beique |

The funders had no role in study design, data collection and interpretation, or the decision to submit the work for publication.

### Author contributions

Jiayun Xu, Investigation, Methodology, Writing - original draft; Mauricio Girardi-Schappo, Conceptualization, Software, Formal analysis, Visualization, Methodology, Writing – review and editing; Jean-Claude Beique, Writing - original draft; André Longtin, Formal analysis, Writing - original draft, Writing – review and editing; Leonard Maler, Conceptualization, Formal analysis, Supervision, Funding acquisition, Validation, Visualization, Methodology, Writing - original draft, Project administration, Writing – review and editing

## Author ORCIDs

Jiayun Xu ⓘ https://orcid.org/0000-0002-9505-9253
Mauricio Girardi-Schappo ⓘ https://orcid.org/0000-0002-9111-4905
Jean-Claude Beique ⓘ https://orcid.org/0000-0001-7278-4906
André Longtin ⓘ https://orcid.org/0000-0003-0678-9893
Leonard Maler ⓘ https://orcid.org/0000-0001-7666-2754

## Ethics

All animal procedures were conducted with the approval of the University of Ottawa's Animal Care Committee and in accordance with guidelines set out by the Canadian Council of Animal Care.

Reviewer #1 (Public review): https://doi.org/10.7554/eLife.95764.4.sa1
Reviewer #3 (Public review): https://doi.org/10.7554/eLife.95764.4.sa2
Author response https://doi.org/10.7554/eLife.95764.4.sa3

## Additional files

### Supplementary files
• MDAR checklist

### Data availability

All data generated and analyzed data in this study is included in this manuscript. All codes and raw data sets are available in https://github.com/neuro-physics/mouse-cogmap (copy archived at *Girardi Schappo, 2024*).

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
