## [Editor Report · eLife Assessment]

This **fundamental** work provides creative and thoughtful analysis of rodent foraging behavior and its dependence on body reference frame-based vs world reference frame-based cues. **Compelling** evidence demonstrates that a robust map, capable of supporting taking novel shortcuts, can be learned primarily if not exclusively based on self-motion cues, which has rarely if ever been reported outside of the human literature. The work, which will be of interest to a broad audience of neuroscientists, provides a rich discussion about the role of the hippocampus in supporting the behavior that should guide future neurophysiological investigations.

---

## [Referee Report · Reviewer #1 (Public review)]

Assessment:

This fundamental work advances our understanding of navigation and path integration in mammals by using a clever behavioral paradigm. The paper provides compelling evidence that mice are able to create and use a cognitive map to find "short cuts" in an environment, using only the location of rewards relative to the point of entry to the environment and path integration, and need not rely on visual landmarks.

Summary:

The authors have designed a novel experimental apparatus called the 'Hidden Food Maze (HFM)' and a beautiful suite of behavioral experiments using this apparatus to investigate the interplay between allothetic and idiothetic cues in navigation. The results presented provide a clear demonstration of the central claim of the paper, namely that mice only need a fixed start location and path integration to develop a cognitive map. The experiments and analyses conducted to test the main claim of the paper -- that the animals have formed a cognitive map -- are conclusive and include many thoughtfully designed control experiments to eliminate alternatives.

Strengths:

The 90 degree rotationally symmetric design and use of 4 distal landmarks and 4 quadrants with their corresponding rotationally equivalent locations (REL) lends itself to teasing apart the influence of path integration and landmark-based navigation in a clever way. The authors use a complete set of experiments and associated controls to show that mice can use a start location and path integration to develop a cognitive map and generate shortcut routes to new locations.

Weaknesses:

There were no major weaknesses identified that were not addressed during revisions.

---

## [Referee Report · Reviewer #3 (Public review)]

Summary:

How is it that animals find learned food locations in their daily life? Do they use landmarks to home in on these learned locations or do they learn a path based on self-motion (turn left, take ten steps forward, turn right, etc.). This study carefully examines this question in a well-designed behavioral apparatus. A key finding is that to support the observed behavior in the hidden food arena, mice appear to not use the distal cues that are present in the environment for performing this task. Removal of such cues did not change the learning rate, for example. In a clever analysis of whether the resulting cognitive map based on self-motion cues could allow a mouse to take a shortcut, it was found that indeed they are. The work nicely shows the evolution of the rodent's learning of the task, and the role of active sensing in the targeted reduction of uncertainty of food location proximal to its expected location.

Strengths:

A convincing demonstration that mice can synthesize a cognitive map for the finding of a static reward using body frame-based cues. Showing that uncertainty of final target location is resolved by an active sensing process of probing holes proximal to the expected location. Showing that changing the position of entry into the arena rotates the anticipated location of the reward in a manner consistent with failure to use distal cues.

Weaknesses:

Weaknesses: The Reviewing Editor felt that previously identified weaknesses from Reviewer #3 were adequately addressed in the final manuscript.

---

## [Author Response]

The following is the authors’ response to the previous reviews.

I have added a paragraph that addresses the issue of how landmarks might be used and why they are not. The suggestions made in the "Weaknesses" paragraph were concise and excellent and have directly incorporated them into my revised manuscript. This text appears on Page 21 and is shown below. I hope that this is what the editors and reviewers were looking.

The requested revision is the second paragraph.

The first paragraph was not written in response to reviews but inspired by a recent paper by Mahdev et al (2024) - https://doi.org/10.1038/s41593-024-01681-9. I had already requested to add this reference and was encouraged to do so by the Editors. The Mahdev et al paper was very surprising in that it showed that path integration is not constant but that its "gain" can be recalibrated by selfmotion signals. I wondered whether this unexpected capacity extended to path integration also recalibrating the cognitive map and thereby generating the shortcutting behavior we observe. I suggested that, at an abstract level, this would correspond to "coordinate transformation" of the cognitive map. I realize that this is entirely speculative. If the Editors feel that it does not add much to the manuscript and that the speculation goes to far, I will remove the first paragraph and re-submit.

Added text. P21 and just before the heading: " Implications for theories of hippocampal representations of spatial maps" There were no other changes made in the paper.

"Path integration uses self-motion signals to update the animal's estimated location on its internal cognitive map. Path integration gain has been shown to be plastic and regulated by landmarks (*52*). Remarkably, a recent study has revealed that path integration gain can also be directly recalibrated by self-motion signals alone (*53*), albeit not as effectively as by landmarks (*52, 53*). An interesting question for future research is whether self-motion signals can also recalibrate the coordinates of a cognitive map. From this perspective, the Target B to Target A shortcut requires a transformation of the cognitive map coordinates so that the start point is now Target B.

Extensive research has shown that external cues can control hippocampal neuron place fields (*11, 12, 54*) and the gain of the path integrator (*52*), making the failure of mice in our study to use such cues puzzling. The failure to use landmarks may be related to our task being low stakes and our pretraining procedure teaching the mouse that such cues are not necessary. Our results may not generalize to more natural conditions where many reliable prominent cues are available, and where there is urgency to find food or water while avoiding predation (*55*). Under these more naturalistic conditions the use of distal cues to rapidly find a food reward is more likely to be observed."